# IDArb: Intrinsic Decomposition for arbitrary number of input views and illuminations

**Zhibing Li**[1], **Tong Wu**[1†], **Jing Tan**[1], **Mengchen Zhang**[2,3], **Jiaqi Wang**[3], **Dahua Lin**[1,3,4†]
[1] The Chinese University of Hong Kong     [2] Zhejiang University
[3] Shanghai AI Laboratory     [4] CPII under InnoHK
{lz022, wt020, dhlin}@ie.cuhk.edu.hk

## Abstract

Capturing geometric and material information from images remains a fundamental challenge in computer vision and graphics. Traditional optimization-based methods often require hours of computational time to reconstruct geometry, material properties, and environmental lighting from dense multi-view inputs, while still struggling with inherent ambiguities between lighting and material. On the other hand, learning-based approaches leverage rich material priors from existing 3D object datasets but face challenges with maintaining multi-view consistency. In this paper, we introduce IDArb, a diffusion-based model designed to perform intrinsic decomposition on an arbitrary number of images under varying illuminations. Our method achieves accurate and multi-view consistent estimation on surface normals and material properties. This is made possible through a novel cross-view, cross-domain attention module and an illumination-augmented, view-adaptive training strategy. Additionally, we introduce ARB-Objaverse, a new dataset that provides large-scale multi-view intrinsic data and renderings under diverse lighting conditions, supporting robust training. Extensive experiments demonstrate that IDArb outperforms state-of-the-art methods both qualitatively and quantitatively. Moreover, our approach facilitates a range of downstream tasks, including single-image relighting, photometric stereo, and 3D reconstruction, highlighting its broad applications in realistic 3D content creation. Project website: https://lizb6626.github.io/IDArb/.

## 1 Introduction

The color we perceive from objects results from a complex interaction between the incident light, the material properties, and the surface geometry of those objects. Recovering these intrinsic properties from captured images is a fundamental challenge in computer vision, enabling a variety of downstream applications, such as relighting (Wimbauer et al., 2022) and photo-realistic 3D content generation (Zhang et al., 2024; Siddiqui et al., 2024). This decomposition process, commonly referred to as *inverse rendering*, is inherently ambiguous and severely under-constrained, particularly when only one or a limited number of observation views are available. For instance, a black pixel could indicate black base color or is the result of lacking incident light.

Existing inverse rendering research can be broadly categorized into two approaches: optimization-based methods and learning-based methods. The former category (e.g. NeRFactor (Zhang et al., 2021b), NVDiffRecMC (Hasselgren et al., 2022), TensoIR (Jin et al., 2023)) typically requires hundreds of multi-view images as input and focuses on optimizing intrinsic properties for each case independently. This approach involves time-consuming iterative optimization, often requiring several hours. Moreover, without incorporating strong priors on material distribution or addressing the inherent ambiguity between lighting and texture, these optimization-based methods frequently converge to sub-optimal solutions. This can lead to unrealistic decompositions, such as embedding lighting effects into intrinsic components, as shown in Fig. 1(b). To address these limitations, learning-based methods aim to extract useful priors from large-scale training datasets and perform fast inference in a feed-forward manner. While many of these approaches focus on single-image decomposition, they tend to produce inconsistent intrinsic properties when applied across multiple

---

†Corresponding authors.

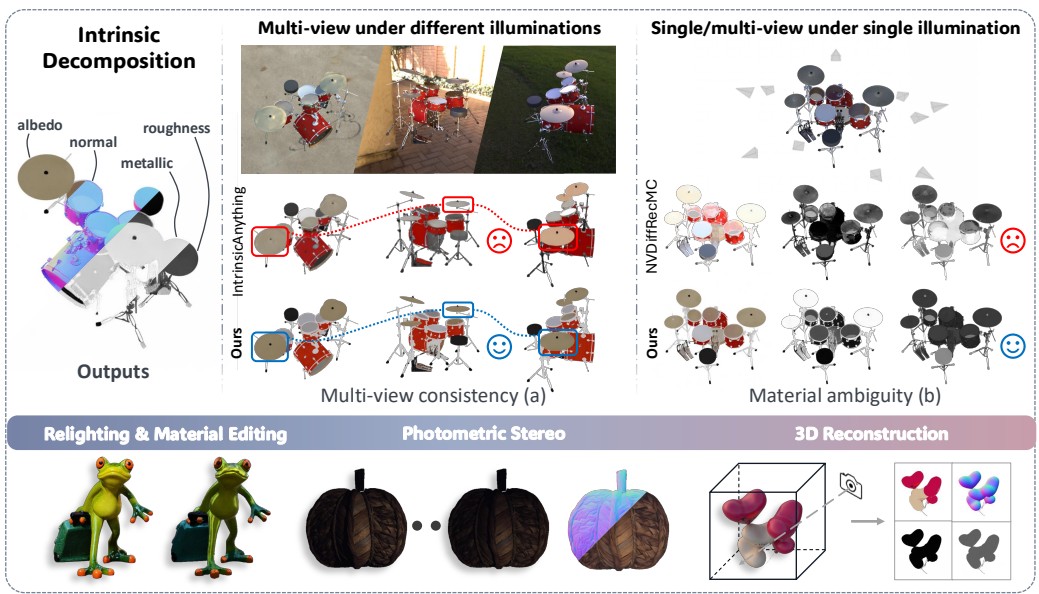

Figure 1: **IDArb tackles intrinsic decomposition for an arbitrary number of views under unconstrained illumination.** Our approach (a) achieves multi-view consistency compared to learning-based methods and (b) effectively disentangles intrinsic components from lighting effects compared to optimization-based methods. Our method enhances a wide range of applications such as image editing, photometric stereo, and 3D reconstruction.

views, as demonstrated in Fig. 1(a). Additionally, single-image models struggle to leverage complementary information from multiple views, making it difficult to resolve material ambiguities, which results in less accurate outcomes in more complex cases.

To mitigate these challenges, we propose IDArb, a model capable of taking an arbitrary number of images captured under unconstrained, varying lighting conditions and predicting corresponding intrinsic components, including albedo, normal, metallic and roughness. Our key contributions are three-fold. **First**, we adopt the cross-view, cross-component attention module from Wonder3D (Long et al., 2023) to fuse information across different views and intrinsic components. This module facilitates holistic understanding of the multi-view correspondence and joint distribution of intrinsic components, enabling consistency across viewpoints and reducing decomposition uncertainty. Despite being trained on fixed number of input views, our model shows the flexibility to decompose an arbitrary number of input images without requiring camera poses. **Second**, to improve performance under complex lighting conditions, we create a custom dataset based on Objaverse (Deitke et al., 2022), namely ARB-Objaverse, which contains 5.7M multi-view RGB images and intrinsic components with varying illumination scenarios for effective training. **Lastly**, we devise a novel and effective illumination-augmented and view-adapted training strategy to achieve robust performance under varying lighting conditions and leverage both multi-view cues and general object material prior for better multi-view and single-view inverse rendering.

We evaluate our model extensively on both synthetic and real data. Our approach significantly outperforms existing learning-based methods (Kocsis et al., 2024; Zeng et al., 2024; Chen et al., 2024) by a large margin, both qualitatively and quantitatively, achieving state-of-the-art results in intrinsic decomposition. Our model offers practical benefits for a range of downstream tasks, including material editing, relighting, and photometric stereo, and it can also serve as a strong prior to improve optimization-based methods by better disentangling lighting effects from intrinsic appearance. We believe that IDArb provides a unified solution across different input regimes in inverse rendering, advancing our ability to understand and model the physical world.

## 2 RELATED WORK

### 2.1 OPTIMIZATION-BASED INVERSE RENDERING

Optimization-based inverse rendering methods aim to jointly reconstruct shape, materials, and lighting from multi-view images. Volumetric representation methods (Boss et al., 2021a; Kuang et al.,

2022; Boss et al., 2021b; Zhang et al., 2021b) extend NeRF (Mildenhall et al., 2020) to model intrinsic appearance and lighting conditions, rendering images using volume rendering techniques. Surface-based representation methods (Zhang et al., 2021a; 2022a;b; Sun et al., 2023; Wu et al., 2023) extract surfaces as signed distance functions (SDFs) (Wang et al., 2021) or differentiable meshes (Munkberg et al., 2022; Hasselgren et al., 2022), apply explicit material models such as Bidirectional Reflectance Distribution Functions (BRDFs) (Nicodemus, 1965), and render images through physics-based procedures. Recent works explore 3D Gaussian representation Kerbl et al. (2023); Gao et al. (2023) for this task, assigning intrinsic attributes to each Gaussian point.

While existing methods effectively simulate global illumination, they often require dense multi-view inputs and can be computationally expensive, especially for complex scenes. In addition, they face the inherent ambiguity between lighting and materials, which can lead to suboptimal solutions, such as incorrectly baked lighting into textures. In contrast, our proposed method offers an efficient solution for inverse rendering in a feed-forward manner. By leveraging well-learned priors from our large-scale, multi-view, multi-lighting dataset, we can significantly mitigate the issue of ambiguity.

## 2.2 LEARNING-BASED INVERSE RENDERING

With advances in deep neural networks, learning-based approaches (Barron & Malik, 2020; Li et al., 2019; Zhu et al., 2022; Bi et al., 2020; Careaga & Aksoy, 2023; Shi et al., 2016) have demonstrated impressive performance in intrinsic decomposition. They typically take a single image as input and decompose intrinsic properties from the input view, such as albedo, specular, and surface normal. Early learning-based methods (Li et al., 2018; Wu et al., 2021; Wimbauer et al., 2022; Sang & Chandraker, 2020; Boss et al., 2020; Yi et al., 2023) handle intrinsic decomposition as a deterministic problem, often leading to over-smoothed details in ambiguous pixels. Recent works (Kocsis et al., 2024; Chen et al., 2024; Zeng et al., 2024) adopt probabilistic distribution modeling with diffusion (Ho et al., 2020), estimating accurate intrinsic components with high-frequency details through a generative formulation. Zeng et al. (2024) presents a unified diffusion framework that addresses both RGB→X (estimating intrinsic properties) and X→RGB (generating realistic images) by training diffusion pipelines on multiple data sources.

These learning-based approaches typically handle inverse rendering in a single-view setting, leading to inconsistent results when applied to multi-view data. Our work extends the feed-forward diffusion pipeline to address the under-explored challenge of multi-view inverse rendering, providing a unified solution for various input types and offering valuable intrinsic priors for downstream applications.

## 2.3 DIFFUSION MODELS FOR OTHER MODALITIES

Denoising Diffusion Probabilistic Models (DDPMs) and their variants (Ho et al., 2020; Rombach et al., 2021; Zhang et al., 2023) have gained significant attention in text-to-image generation, yielding promising results across various applications. Researchers have also explored adapting diffusion models to different output modalities such as normal (Fu et al., 2024), depth (Ke et al., 2024) and novel view images (Liu et al., 2023; 2024b; Kong et al., 2024). To generate multiple modality simultaneously, Wonder3D (Long et al., 2023) introduces additional cross-domain attention modules into diffusion model that generates multi-view normal maps and corresponding color images. We extend this concept to intrinsic decomposition by splitting the intrinsic components into three triplets and modeling their joint distribution. By leveraging pre-trained diffusion models, which capture rich structural, semantic, and material knowledge, we can overcome data limitations and ensure generalization to real-world scenarios, even when the models are trained on synthetic data.

## 3 METHOD

IDArb is a diffusion-based model for intrinsic decomposition that can handle an arbitrary number of input views and varying lighting conditions. We begin by outlining the problem statement in Section 3.1. Then, in Section 3.2, we describe the construction of our custom dataset tailored to this task. Finally, we discuss the model architecture and training strategy in Sec. 3.3. An overview of IDArb is provided in Fig. 2.

## 3.1 PROBLEM STATEMENT

We frame intrinsic decomposition as a conditional generation problem:

$$\mathbf{X}_{1:N} \sim p(\mathbf{X}_{1:N}|\mathbf{I}_{1:N}). \tag{1}$$

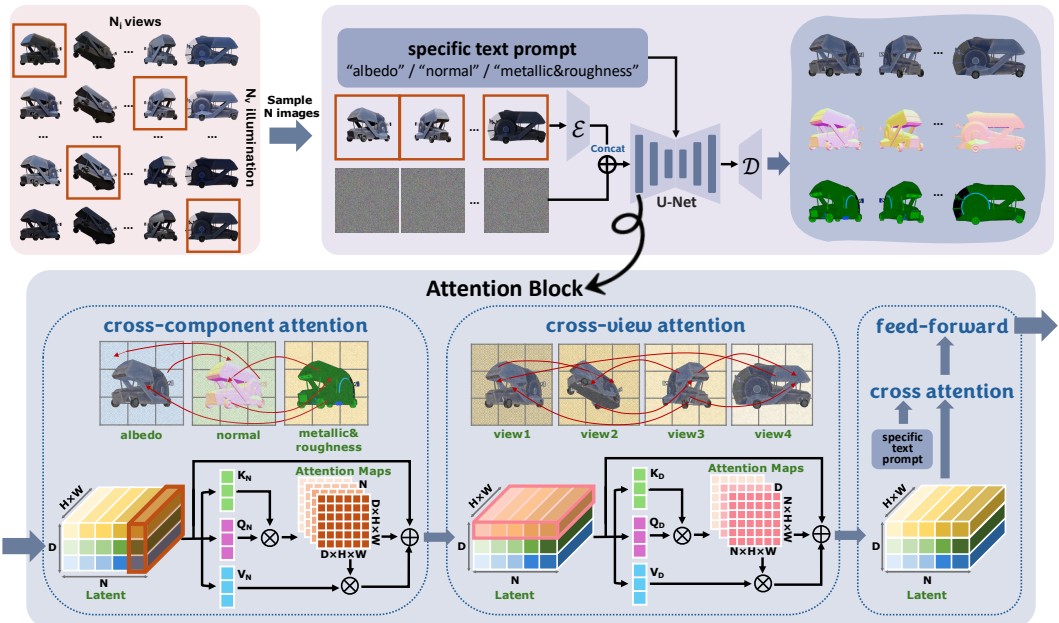

Figure 2: **Top: Overview of IDArb. Bottom: Illustration of the attention block within the UNet.** Our training batch consists of $N$ input images, sampled from $N_v$ viewpoints and $N_i$ illuminations. The latent vector for each image is concatenated with Gaussian noise for denoising. Intrinsic components are divided into three triplets ($D$=3): Albedo, Normal and Metallic&Roughness. Specific text prompts are used to guide the model toward different intrinsic components. For attention block inside UNet, we introduce cross-component and cross-view attention module into it, where attention is applied across components and views, facilitating global information exchange.

Here, $N \in \mathbb{N}$ denotes the number of input views; $\mathbf{I}_{1:N}$ denotes input RGB images, and $\mathbf{X}_{1:N}$ represents the intrinsic components of each view. We model $\mathbf{X}$ using the simplified Disney BRDF parameterization (Burley & Studios, 2012; Karis & Games, 2013), which includes albedo $\mathbf{A} \in \mathbb{R}^{H \times W \times 3}$, roughness $\mathbf{R} \in \mathbb{R}^{H \times W \times 1}$, metallic $\mathbf{M} \in \mathbb{R}^{H \times W \times 1}$ and surface normal $\mathbf{N} \in \mathbb{R}^{H \times W \times 3}$. The number of input images $N$ can take on an arbitrary value from one to many, and the input images can be rendered under arbitrary unconstrained illuminations during both training and inference.

## 3.2 ARB-OBJAVERSE DATASET

Obtaining ground truth data for intrinsic decomposition in real-world settings is both time-consuming and technically challenging. To overcome this, we rely on synthetic data for training. Ideally, a suitable dataset should feature large-scale, diverse objects rendered under multiple lighting conditions. However, existing datasets have notable limitations. For example, G-Objaverse (Qiu et al., 2024) employs a single, low-contrast lighting setup, while ABO (Collins et al., 2022) is restricted to household items, suffering from a lack of diversity among the objects.

To address these shortcomings, we develop a custom dataset, Arb-Objaverse. We select 68k 3D models from Objaverse Deitke et al. (2022), and filter out low-quality and texture-less cases. For each object, we render 12 views, using the Cycles render engine from Blender. For each viewpoint, we render 7 images under different lighting conditions. Six images are illuminated by randomly sampled high-dynamic range (HDR) environment maps from Poly Haven, which offers a collection of 718 varied environment maps. The last image is illuminated by two point light sources randomly positioned on a surrounding shell. Our Arb-Objaverse dataset ends up with 5.7 million rendered RGB images along with their intrinsic components. For training, we further enhance the variability by combining this dataset with G-Objaverse and ABO. Fig. 3 offers a visualization and comparison among these datasets.

---

https://www.blender.org/
https://polyhaven.com/

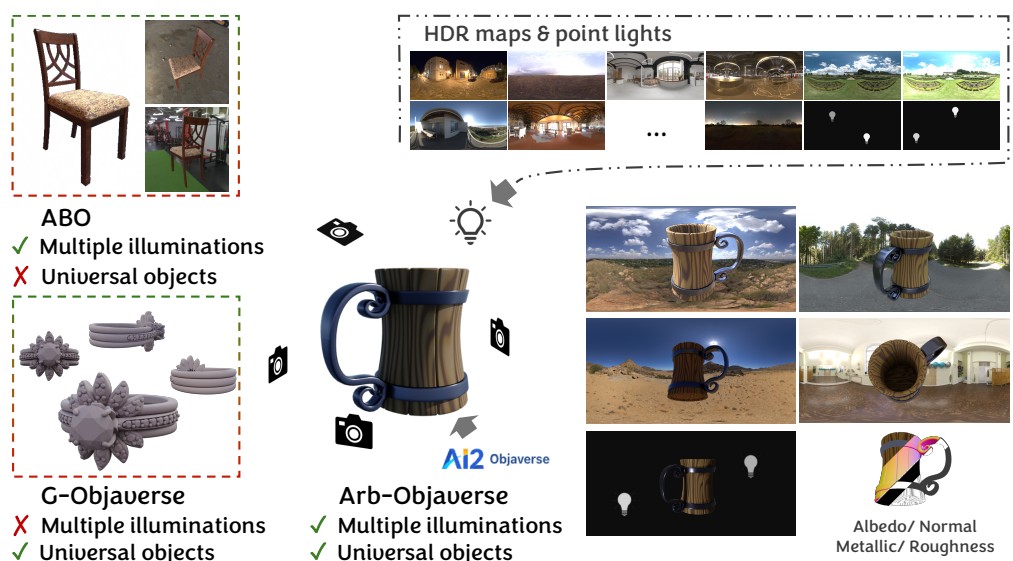

Figure 3: **Overview of the Arb-Objaverse dataset.** Our custom dataset features a diverse collection of objects rendered under various lighting conditions, accompanied by their intrinsic components.

### 3.3 ARCHITECTURE AND TRAINING

Given an arbitrary number of views from single to multi-view images, IDArb generates multi-view consistent intrinsic maps under unconstrained illumination using a text-guided diffusion model. We base our model on the pre-trained Stable Diffusion (SD) (Rombach et al., 2021) model to capitalize on its robust prior knowledge from RGB domain. Different from the 3-channel RGB images, intrinsic components possess higher channel dimensions and cannot be directly processed by original SD model. To repurpose the VAE in original SD for new intrinsic modalities, we divide intrinsic components $\mathbf{X}$ into three triplets: albedo $\mathbf{A}$, normal $\mathbf{N}$ and $\mathbf{B} = [\mathbf{M}, \mathbf{R}, \mathbf{0}]$, where $\mathbf{M}$ is metallic, $\mathbf{R}$ is roughness and $\mathbf{0}$ is left unused. Each triplet latent is channel-concatenated with the Gaussian noise for denoising. Specific text prompts for each triplet, i.e., 'albedo', 'normal', 'metallic&roughness', are devised to indicate denoising targets.

**Cross-view Cross-component Attention.** In real-world scenarios, users may capture multiple images of an object, making it essential for the model to handle an arbitrary number of input views and ensure consistent results across all views. It is also crucial for 3D reconstruction to have these consistent decomposition results as material guidance. To address this, we propose cross-view attention module within the original attention block of UNet. As shown in Fig. 2, we concatenate input features from each view, enabling the attention operation to be performed across views. This allows the model to leverage multi-view information to reduce ambiguity and enforce consistency across different viewpoints.

The reflected color results from the interplay between incident light, material properties, and the surface shape. For instance, a convex shape with a dark color increases the likelihood of a dark albedo. To better capture these relationships, we propose to model the joint distribution of intrinsic components rather than predicting them separately. Inspired by Wonder3D (Long et al., 2023) and GeoWizard (Fu et al., 2024), we adopt cross-component attention via repurposing the vanilla self-attention module to fuse global interactions between different intrinsic components. As demonstrated in Sec. 4.3, exchanging information between components effectively reduces decomposition uncertainty, especially for roughness and metallic.

**Illumination-Augmented and View-Adapted Training.** Multi-view images captured in uncontrolled environments often experience varying lighting conditions, making it essential for algorithms to handle such differences effectively. To address this, we propose an illumination-robust data augmentation strategy, where multi-view images are sampled from various lighting conditions during training. These conditions include a range of setups, such as uniform ambient light, HDR environment maps, and point light sources. At each training step, given $N_v$ views and $N_i$ illumination variations for each instance in the dataset, we randomly sample $N$ images as input. This allows us to

simulate complex input scenarios, including same-view-different-illumination, different-view-same-illumination, and different-view-different-illumination, thus enhancing the diversity of the training data. As a result, our model learns to distinguish different lighting conditions without the need for manually crafted modules, effectively leveraging photometric cues from multi-light captures to achieve robust intrinsic decomposition. It also shows superior generalization capability to handle unseen lighting conditions at inference time.

However, training with fixed $N$ input images leads to downgraded performance when only one view is given (as shown in Sec. 4.3). We suppose that this may be because multi-view training guides the model to focus more on cross-view information to infer intrinsic information, while single-image decomposition requires the learning of general object material priors. To overcome this, we introduce a view-adapted training strategy, that swaps between multi-input and single-image settings. By incorporating this approach, our model gains robust generalization capability with an arbitrary number of input views.

**Noise Scheduler.** The original SD model uses the scaled linear noise scheduler, which prioritizes generating high-frequency details and allocates fewer steps to low-frequency structures. However, this approach limits model's performance in intrinsic decomposition task, as the structure of intrinsic components, particularly metallic $\mathbf{M}$ and roughness $\mathbf{R}$, differs significantly from input RGB images. Inspired by Shi et al. (2023), we shift the noise scheduler toward higher noise levels. As shown in Sec. 4.3, increasing the number of high-noise steps significantly improves the prediction of metallic and roughness components.

## 4 EXPERIMENTS

### 4.1 EXPERIMENTAL SETUP

**Implementation Details.** We finetune the UNet from the pretrained Stable Diffusion with the zero terminal SNR schedule (Lin et al., 2024). We utilize the v-prediction as training objective and the AdamW optimizer with a learning rate of $1 \times 10^{-4}$. The model is trained on downsampled $256 \times 256$ resolution over $80,000$ steps. During training, the number of input images $N$ is randomly set to 3 or 1 per object. The entire training procedure takes approximately 4 days on a cluster of 16 Nvidia Tesla A100 GPUs.

**Baselines.** We compare our method with two recent diffusion-based approaches: IID (Kocsis et al., 2024) and RGB↔X (Zeng et al., 2024). Since RGB↔X is not yet publicly available, we re-implemented it and trained the model on our training dataset. Additionally, we include IntrinsicAnything (Chen et al., 2024) for albedo comparison and GeoWizard (Fu et al., 2024) for normal comparison. We evaluate our model in two settings: (1) single-view setting, where each input image is processed independently, and (2) multi-view setting, where intrinsic components are jointly estimated from multiple views of each object.

**Metrics.** For albedo evaluation, we use Peak Signal-to-Noise Ratio (PSNR) and Structural Similarity Index Measure (SSIM) (Wang et al., 2004). Since albedo is defined up to a scale factor, we apply a scale-invariant PSNR metric by rescaling the predicted albedo as $A' = \mathrm{argmin}_{\alpha}||A - \alpha \hat{A}||^2 \hat{A}$. For surface normals, we measure Cosine Similarity. Mean Squared Error (MSE) is used to evaluate metallic and roughness components.

**Evaluation Dataset.** We evaluate the effectiveness and generalization capability of our model on both synthetic and real-world datasets. For synthetic data, we sample 441 objects from Arb-Objaverse and G-Objaverse, selecting four viewpoints for each object. For real-world data, we collect a set of images from Pixabay. All evaluations are conducted at a resolution of $512 \times 512$.

### 4.2 EXPERIMENTAL RESULTS

**Results on Synthetic Data.** We present quantitative results in Tab. 1, where our method consistently achieves the highest accuracy across all metrics. Fig. 4 displays a visual comparison of our method in single-view setting against baseline methods.

---

https://pixabay.com/

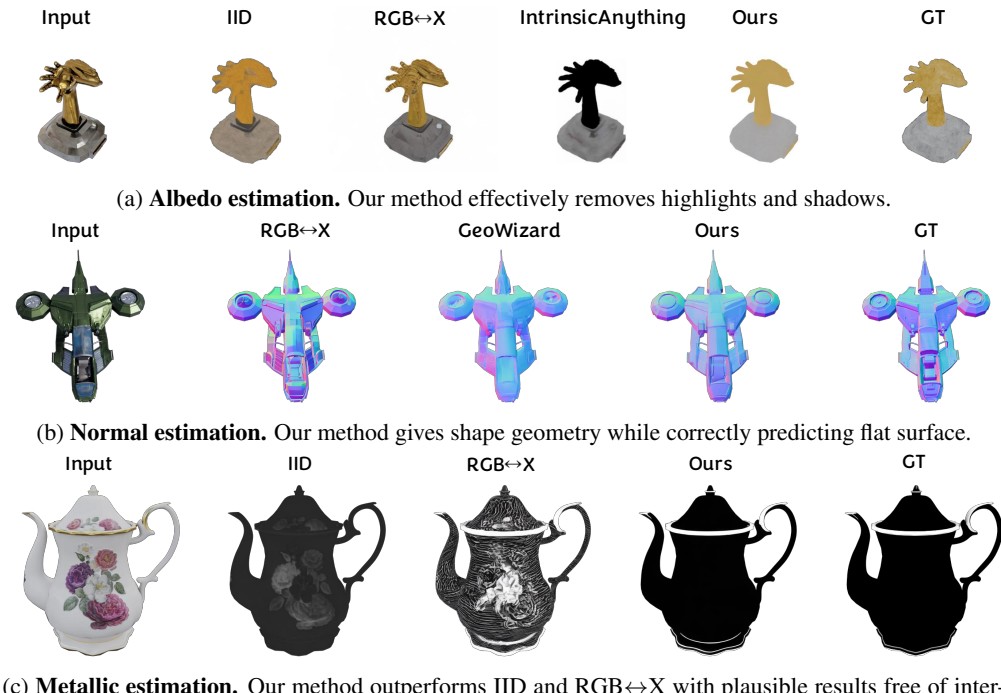

(a) **Albedo estimation.** Our method effectively removes highlights and shadows.

(b) **Normal estimation.** Our method gives shape geometry while correctly predicting flat surface.

(c) **Metallic estimation.** Our method outperforms IID and RGB↔X with plausible results free of interference from texture patterns and lighting.

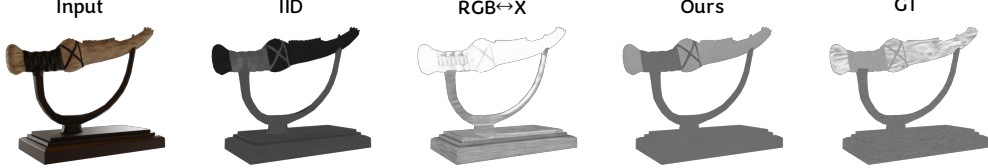

(d) **Roughness estimation.** Our method outperforms IID and RGB↔X with plausible results free of interference from texture patterns and lighting.

Figure 4: **Qualitative comparison on synthetic data.** IDArb demonstrates superior intrinsic estimation compared to all other methods.

Table 1: **Quantitative evaluation of IDArb against baselines.** IDArb consistently achieves the best results among all albedo, normal, metallic and roughness metrics.

|  | Albedo | | Normal | Metallic | Roughness |
|---|---|---|---|---|---|
|  | SSIM↑ | PSNR↑ | Cosine Similarity ↑ | MSE ↓ | MSE ↓ |
| IID | 0.901 | 27.35 | - | 0.192 | 0.131 |
| RGB↔X | 0.902 | 28.09 | 0.834 | 0.162 | 0.347 |
| IntrinsicAnything | 0.901 | 28.17 | - | - | - |
| GeoWizard | - | - | 0.871 | - | - |
| Ours(single) | 0.935 | 32.79 | 0.928 | 0.037 | 0.058 |
| Ours(multi) | **0.937** | **33.62** | **0.941** | **0.016** | **0.033** |

For albedo estimation (Fig. 4a), our method effectively removes highlights and shadows, whereas IID and RGB↔X tend to retain lighting effects in the albedo, and IntrinsicAnything produces unrealistic results for metallic surfaces. In normal estimation (Fig. 4b), our method provides sharp and accurate geometry, while RGB↔X suffers from interference of object textures, and GeoWizard shows blurred details since it evaluates a number of samples and takes their mean. For metallic and roughness estimation (Fig. 4c and Fig. 4d), our method delivers more plausible results, eliminating interference from texture patterns and lighting. Additionally, we observe that incorporating multi-view inputs significantly enhances metallic and roughness predictions, as they provide additional information to resolve material ambiguities.

**Results on Real-world Data.** We present qualitative results on real-world data in Fig. 5 and compare our method with IntrinsicAnything for albedo estimation. IntrinsicAnything predicts overly dark albedo for metallic objects and produces blurry details (such as the toy's mouth in the third row),

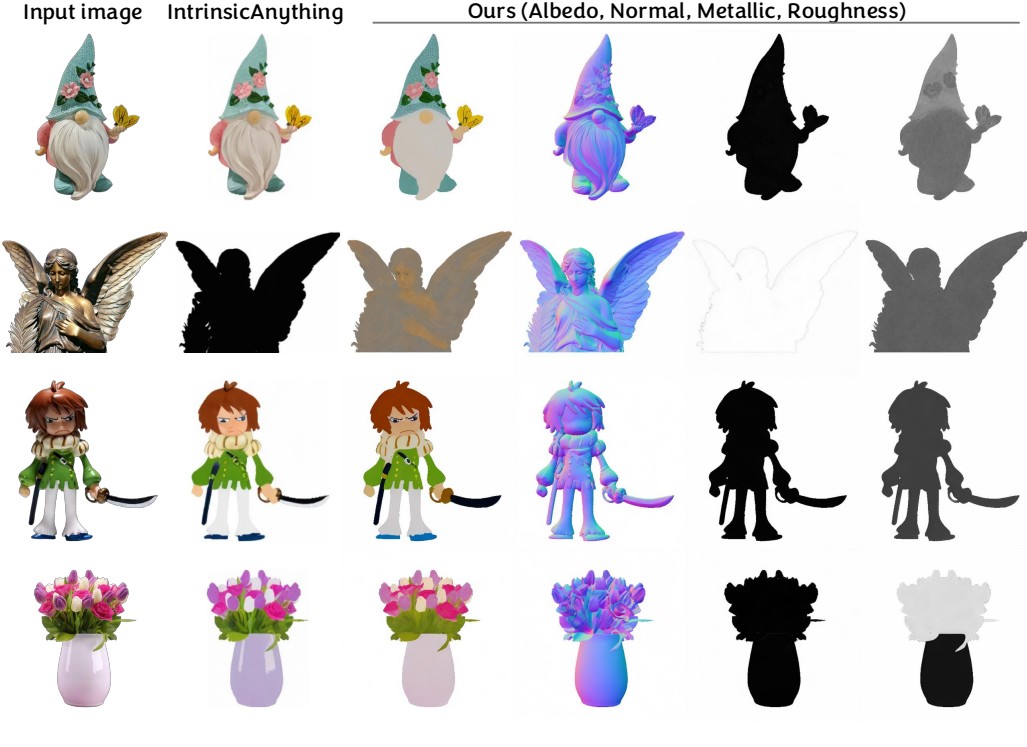

Figure 5: **Qualitative comparison on real-world data.**

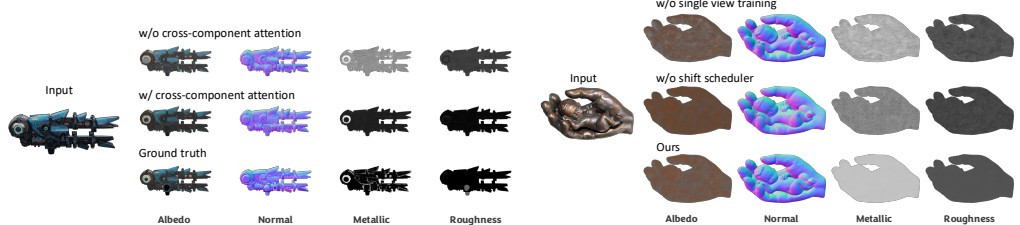

(a) Ablation on cross-component attention.          (b) Ablation on training strategy

Figure 6: **Ablative studies** on **(a) cross-component attention** and **(b) training strategy.**

leading to a loss of fidelity. In contrast, our model generates accurate and convincing decompositions with preserved details. Despite being trained on synthetic data, IDArb generalizes well to real-world images. Additionally, we conduct experiments on standard benchmarks, MIT-Intrinsic (Grosse et al., 2009) and Stanford-ORB (Kuang et al., 2023), with results presented in Appendix. D.

## 4.3 ANALYSIS AND ABLATIVE STUDY

**Ablation on Cross-component Attention.** To assess the effect of cross-component attention, we also trained our model without cross-component attention mechanism for comparison. As shown in Fig. 6a, exchanging information between different intrinsic components helps reduce material ambiguity, particularly for metallic and roughness, which are prone to uncertainty.

**Ablation on Training Strategy.** Fig. 6b shows ablative studies on multi-single view interleaved training strategy and the noise scheduler. Training exclusively on multi-view inputs leads to performance degradation for single-image inputs, as these two settings emphasize different capabilities of the model, as discussed in Sec. 3.3. Additionally, shifting noise scheduler towards high noise level helps the model better adapt to intrinsic domains.

**Analysis of Viewpoints and Lighting Effects.** We analysis the effects of the number of viewpoints and lighting conditions on our custom dataset. We evaluate our model with 1, 2, 4, 8, and 12 viewpoints under 1, 2 and 3 lighting conditions. As shown in Fig. 7, increasing the number of viewpoints

Figure 7: **Effects of number of viewpoints and lighting conditions.** We find increasing the number of viewpoints and the lighting conditions generally improves decomposition performance.

Table 2: **Quantitative results for photometric stereo on NeRFactor.** We evaluate performance using 2, 4, and 8 OLAT images, and achieve the best performance among all compared methods.

| # OLAT Images | 2 | | 4 | | 8 | |
|---|---|---|---|---|---|---|
| Methods | Albedo↑ | Normal↑ | Albedo↑ | Normal↑ | Albedo↑ | Normal↑ |
| IID | 22.23 | - | 22.40 | - | 22.86 | - |
| RGB ↔X | 21.29 | 0.71 | 22.08 | 0.77 | 23.29 | 0.81 |
| SDM-UniPS | 22.95 | 0.74 | 23.20 | 0.76 | 23.37 | 0.81 |
| Ours | **23.50** | **0.83** | **23.64** | **0.84** | **25.15** | **0.85** |

or lights generally improves prediction accuracy. For metallic and roughness predictions, multi-light captures are particularly effective in disentangling these components from lighting effects. Empirically, performance gains from adding more viewpoints diminish beyond eight viewpoints. Further details are provided in Appendix. B.

**More Results.** Additional **multi-view input results** are provided in Appendix. E and supplementary video. For **more real-world data results**, please refer to Appendix. D.

## 4.4 APPLICATIONS

IDArb offers valuable intrinsic priors for various downstream applications. Here, we demonstrate the model's ability in handling single-image relighting and material editing, and photometric stereo problems. Additionally, we show that our generated intrinsic decompositions enhance the results of optimization-based inverse rendering.

**Single-image Relighting and Material Editing.** Once high-quality intrinsic components are obtained, our method enables relighting of captured images under novel illumination. Additionally, we can optimize the lighting in the original scene and perform material editing. Specifically, we represent environment lighting as a cube map and adopt a differentiable split-sum approximation in NVDiffRec (Munkberg et al., 2022) to optimize its parameters. Fig. 8 showcases our relighting and material editing results.

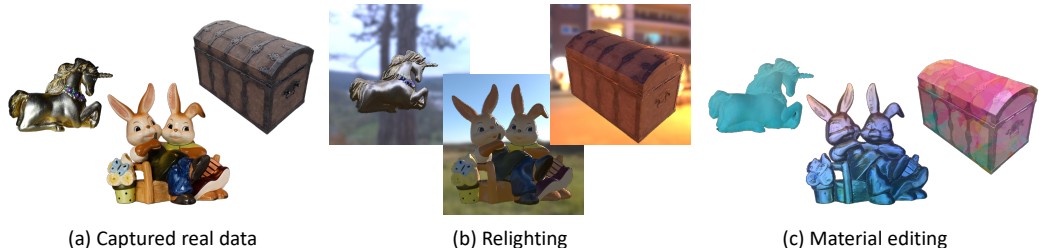

(a) Captured real data                (b) Relighting                (c) Material editing

Figure 8: **Relighting and material editing results.** From in-the-wild captures (a), our model allows for relighting under novel illumination (b) and material property modifications (c).

**Photometric stereo.** Photometric stereo is a long-standing challenge in computer vision, aiming to deducing the surface normal and albedo from images captured under varying lighting conditions with a fixed camera. We evaluate our method under the harsh One-Light-At-a-Time (OLAT) condition, where each image is illuminated by a single point light source without ambient illumination, leading to hard cast shadows. We additionally include SDM-UniPS (Ikehata, 2023) for comparison, which is specifically designed and trained for this task. We conduct experiments on the real-world OpenIllumination dataset (Liu et al., 2024a) and the synthetic NeRFactor dataset (Zhang et al.,

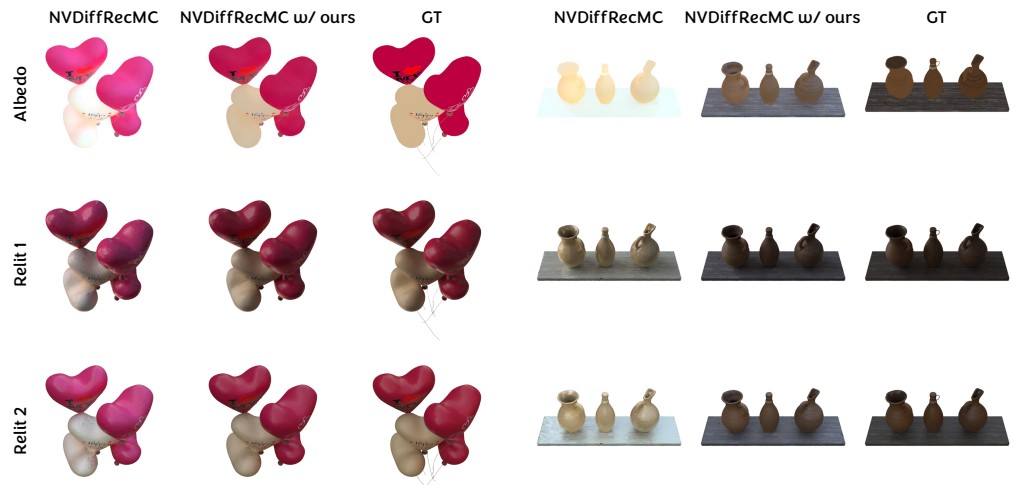

Figure 9: **Optimization-based inverse rendering results.** Our method guides NVDiffecMC generate more plausible material results.

Table 3: **Ablation on IDArb pseudo labels for optimization-based inverse rendering** on NeRFactor and Synthetic4Relight datasets.

| | Nerfactor | | | Synthetic4Relight | | | |
| --- | --- | --- | --- | --- | --- | --- | --- |
| | Albedo (raw) | Albedo (scaled) | Relighting | Albedo (raw) | Albedo (scaled) | Relighting | Roughness |
| NVDiffRecMC | 17.89 | 25.88 | 22.65 | 17.03 | 29.64 | 24.05 | 0.046 |
| NVDiffRecMC w/ Ours | **20.90** | **26.61** | **27.20** | **26.42** | **30.73** | **31.01** | **0.014** |

2021b). Quantitative results on NeRFactor are summarized in Tab. 2, and qualitative results are present in Appendix. C. Although our model is not explicitly trained for this setting, it still delivers reasonable estimates, particularly when the number of input images is limited.

**Optimization-based Inverse Rendering.** Our method can be used as a prior to enhance optimization-based inverse rendering techniques. Specifically, we decompose each training image into its corresponding intrinsic components and treat these components as pseudo-material labels. We adopt NVDiffRecMC (Hasselgren et al., 2022) as the codebase for our experiments, as it employs the same PBR material model as our method. During each iteration, we introduce an additional L2 regularization term between the intrinsic components predicted by NVDiffRecMC and those predicted by our method to ensure physical plausibility. Tab. 3 presents material estimation and relighting results on these dataset. As illustrated in Fig. 9, our method significantly mitigates the color-shifting issue in the reconstructed albedo from NVDiffRecMC, leading to improved results in relighting tasks.

## 5 CONCLUSION

In this paper, we present IDArb, that solves intrinsic decomposition via a feed-forward diffusion pipeline. Our method can process arbitrary images captured under unknown and varying illuminations and estimate consistent intrinsic components, including albedo, normal, metallic and roughness. The cross-component attention module and illumination-augmented training further enhance our model's ability to reduce ambiguity, fostering more robust inverse rendering under complex, high-contrast lighting conditions.

**Limitations and Discussions.** While our method demonstrates strong generalization capabilities on real-world data, it faces challenges in accurately predicting material maps for intricate objects, such as corroded bronze statues with spatially varying metallic and roughness properties due to corrosion levels. Given that most synthetic data employ global metallic and roughness values, our method may oversimplify estimations for complex real-world objects. Future research directions could involve incorporating real data through unsupervised techniques. Moreover, the current implementation of cross-view attention concatenates all input views, leading to a complexity of $O(N^2)$ and posing difficulties in handling dense input views with high resolutions. Future investigations could explore more efficient cross-view attention mechanism. Further discussion on failure cases can be found in Appendix F.

ACKNOWLEDGEMENT

This project is funded in part by Shanghai Artificial lntelligence Laboratory, the National Key R&D Program of China (2022ZD0160201), the Centre for Perceptual and Interactive Intelligence (CPII) Ltd under the Innovation and Technology Commission (ITC)'s InnoHK. Dahua Lin is a PI of CPII under the InnoHK.

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

## A  PRELIMINARY

### A.1  IMAGE DIFFUSION MODEL

In Denoising Diffusion Probabilistic Models (DDPM) (Ho et al., 2020), a forward diffusion process is defined, gradually introducing small amounts of Gaussian noise to the sample at each timestep, represented by $q(\mathbf{x}_t|\mathbf{x}_{t-1}) = \mathcal{N}(\mathbf{x}_t; \sqrt{1 - \beta_t}\mathbf{x}_{t-1}, \beta_t\mathbf{I})$, where $t$ represents the timestep and $\beta$ acts as the variance scheduler. To recover samples from the random noise, DDPM learns to model the reverse diffusion process as $p_\theta(\mathbf{x}_{t-1}|\mathbf{x}_t) = \mathcal{N}(\mathbf{x}_{t-1}; \mu_\theta(\mathbf{x}, t), \Sigma_\theta(\mathbf{x}_t, t))$ and construct $\mathbf{x}_0$ through iterative denoising.

Stable Diffusion (SD) (Rombach et al., 2021) employ an encoder $\mathcal{E}$ to compress the input image $\mathbf{x} \in \mathbb{R}^{H \times W \times 3}$ into a latent vector $\mathbf{z} \in \mathbb{R}^{H/8 \times W/8 \times 4}$ before performing the diffusion process in the latent space. Following denoising, the latent representation is then converted back to pixel space through a decoder $\hat{x} = \mathcal{D}(\mathbf{z}_0)$.

For conditional generation, the training objective of Stable Diffusion (SD) is formulated as:

$$L := \mathbb{E}_{\mathcal{E}(\mathbf{x}), y, \epsilon \sim \mathcal{N}(0,1), t}[||\epsilon - \epsilon_\theta(\mathbf{z}_t, t, \tau_\theta(y))||_2^2], \tag{2}$$

where $t$ is uniformly sampled from $\{1, ..., T\}$, $\tau_\theta(y)$ represents the encoding of the condition $y$ and $\epsilon_\theta$ is implemented as a UNet.

## A.2 INTRINSIC COMPONENTS FORMATION

Our image formation is based on the classic rendering equation (Kajiya, 1986) to ensure physical correctness. For a point $\mathbf{x}$ with surface normal $\mathbf{n}$, the incident light intensity at this point is denoted as $L_i(\omega_i; x)$, where $\omega_i$ represents the incident light direction. The Bidirectional Reflectance Distribution Function (BRDF) (Nicodemus, 1965), denoted as $f_r(\omega_o, \omega_i; x)$, describes the reflectance properties of the material when viewed from direction $\omega_o$. The observed light intensity $L_o(\omega_0; x)$ is calculated over the hemisphere $\Omega = \{\omega_i : \omega_i \cdot n > 0\}$ as follows:

$$L_o(\omega_o; x) = \int_{\Omega} L_i(\omega_i; x) f_r(\omega_o, \omega_i; x)(\omega_i \cdot n) d\omega_i. \tag{3}$$

In our approach, we aim to recover the object's **surface normal** and BRDF material from the observed color on the left-hand side of Eq. 3, which are independent of illumination and view direction. We adopt the Disney Basecolor-Metallic model(Burley & Studios, 2012) for BRDF parametrization, which comprises the following components: **albedo**, representing the base color; **roughness**, controlling the diffuse and specular response; and **metallic**, governing the specular reflection.

Specifically, given a single RGB image $\mathbf{I} \in \mathbb{R}^{H \times W \times 3}$, we aim to jointly estimate the surface normal $\mathbf{N} \in \mathbb{R}^{H \times W \times 3}$, albedo $\mathbf{A} \in \mathbb{R}^{H \times W \times 3}$, roughness $\mathbf{R} \in \mathbb{R}^{H \times W \times 1}$ and metallic $\mathbf{M} \in \mathbb{R}^{H \times W \times 1}$.

## B DETAILS ABOUT THE EFFECTS OF VIEWPOINTS AND LIGHTING

We present the numerical performance results across varying numbers of viewpoints (# V) and lighting conditions (# L), as shown in Tab. 4 to 7.

Table 4: **Albedo Performance** ↑ across different numbers of viewpoints (# V) and lightings (# L).

| # L \ # V | 1 | 2 | 4 | 8 | 12 |
|---|---|---|---|---|---|
| 1 | 29.16 | 28.72 | 30.12 | 30.49 | 30.77 |
| 2 | 29.96 | 30.26 | 30.96 | 31.13 | 31.26 |
| 3 | 30.25 | 30.73 | 31.16 | 31.33 | 31.40 |

Table 5: **Normal Performance** ↑ across different numbers of viewpoints (# V) and lightings (# L).

| # L \ # V | 1 | 2 | 4 | 8 | 12 |
|---|---|---|---|---|---|
| 1 | 0.909 | 0.910 | 0.925 | 0.930 | 0.932 |
| 2 | 0.922 | 0.927 | 0.930 | 0.933 | 0.934 |
| 3 | 0.926 | 0.931 | 0.931 | 0.934 | 0.935 |

Table 6: **Metallic Performance** ↓ across different numbers of viewpoints (# V) and lightings (# L).

| # L \ # V | 1 | 2 | 4 | 8 | 12 |
|---|---|---|---|---|---|
| 1 | 0.105 | 0.116 | 0.068 | 0.059 | 0.050 |
| 2 | 0.061 | 0.068 | 0.047 | 0.044 | 0.042 |
| 3 | 0.061 | 0.056 | 0.048 | 0.045 | 0.040 |

Table 7: **Roughness Performance** ↓ across different numbers of viewpoints (# V) and lightings (# L).

| # L \ # V | 1 | 2 | 4 | 8 | 12 |
|---|---|---|---|---|---|
| 1 | 0.049 | 0.050 | 0.024 | 0.019 | 0.021 |
| 2 | 0.043 | 0.026 | 0.019 | 0.016 | 0.015 |
| 3 | 0.031 | 0.022 | 0.016 | 0.014 | 0.013 |

## C ADDITIONAL RESULTS ON PHOTOMETRIC STEREO

We present qualitative results of photometric stereo in Fig. 10.

## D ADDITIONAL RESULTS ON REAL-WORLD DATA

We evaluate our method on two real-world benchmarks: MIT-Intrinsic (Grosse et al., 2009) and Stanford-ORB (Kuang et al., 2023). For MIT-Intrinsic, we compared our albedo estimation results with IntrinsicAnything (Chen et al., 2024), as shown in Tab. 8 and Fig. 11. For Stanford-ORB, we presented results for normal estimation, albedo estimation, and re-rendering, comparing our method with StableNormal (Ye et al., 2024) and IntrinsicNeRF (Ye et al., 2023), as shown in Tab. 9. For the re-rendering evaluation, we utilized the ground truth environment maps to render our decomposition results and compared them with the original images.

Additionally, we present qualitative results on real-world data from the Internet in Fig. 12 and Fig. 13.

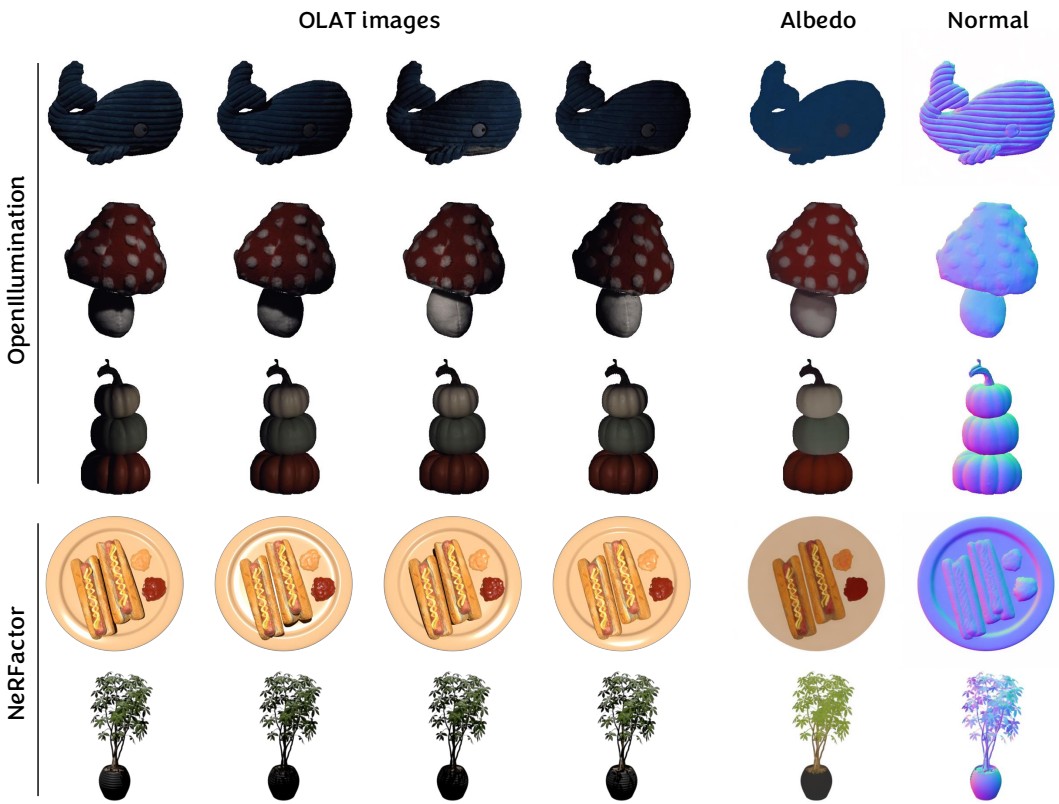

Figure 10: **Photometric stereo results** using 4 OLAT images in OpenIllumination and NeRFactor.

Table 8: **Quantitative comparisons on MIT-Intrinsic.**

|  | SSIM↑ | PSNR↑ | LPIPS↓ |
|---|---|---|---|
| Ours | 0.876 | **27.98** | **0.117** |
| IntrinsicAnything | **0.896** | 25.66 | 0.150 |

Table 9: **Quantitative comparisons on Stanford-ORB.**

|  | Normal | Albedo | | | Re-rendering | | | |
|---|---|---|---|---|---|---|---|---|
|  | Cosine Distance↓ | SSIM↑ | PSNR↑ | LPIPS ↓ | PSNR-H↑ | PSNR-L↑ | SSIM↑ | LPIPS ↓ |
| Ours(single) | 0.041 | 0.978 | 41.30 | 0.039 | 24.11 | 31.28 | 0.969 | 0.024 |
| Ours(multi) | **0.029** | 0.978 | **41.46** | **0.038** | **24.36** | **31.43** | **0.970** | **0.024** |
| StableNormal | 0.038 |  |  |  |  |  |  |  |
| IntrinsicNeRF |  | **0.981** | 39.31 | 0.048 |  |  |  |  |

# E ADDITIONAL RESULTS ON MULTI-VIEW INPUTS

We present additional results on multi-view input in Fig. 14.

# F FAILURE CASES

Several failure cases are illustrated in Fig. 16. First, our model struggles with outdoor scenes, as it is primarily trained on object-centric data. While the model exhibits some generalization capability, its performance degrades in these scenarios. Second, when the model is faced with text, the decomposition fails to recover the correct text structures. Finally, in the third row, the model produces overly simplified outputs in certain cases, failing to preserve subtle material details, such as the metallic

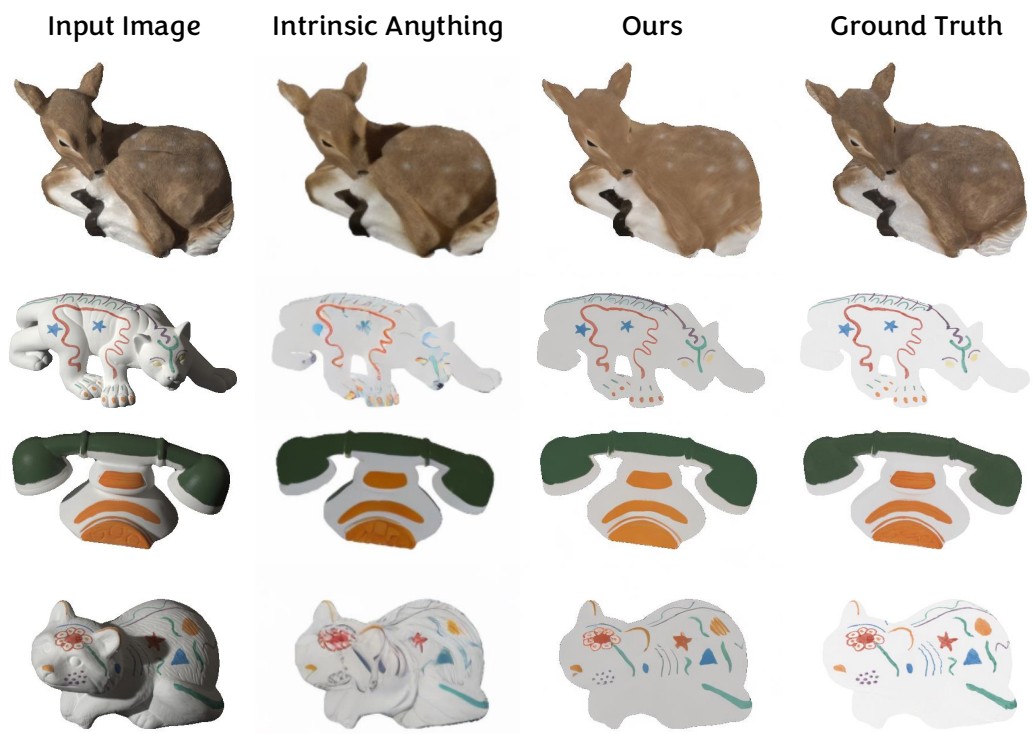

Figure 11: **Qualitative comparison on MIT-Intrinsic (Grosse et al., 2009) with Intrinsi-cAnything (Chen et al., 2024).** Input image and ground truth have been contrast-adjusted for better visibility.

features of a telephone. This issue arises from the synthetic training data, which often contains simpler material variations, leading the model to overly simplify fine-grained material properties.

## G  GENERALIZATION TO SCENE-LEVEL DATA

Despite not being explicitly trained on such datasets, our model demonstrates generalization ability in outdoor and indoor scenes. We provide qualitative results in Fig. 17, Fig. 18 and Fig. 19.

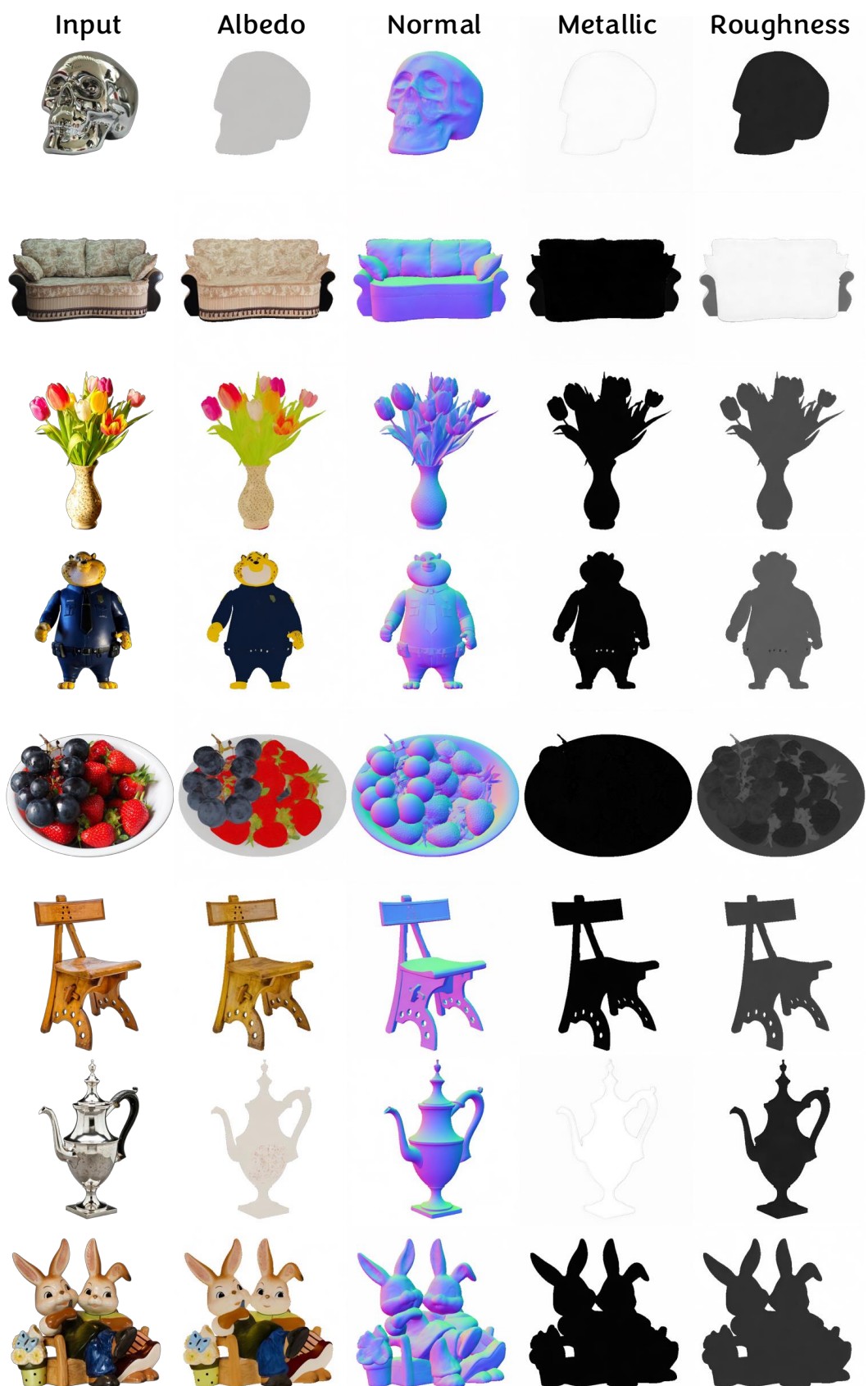

Figure 12: More results on real-world data.

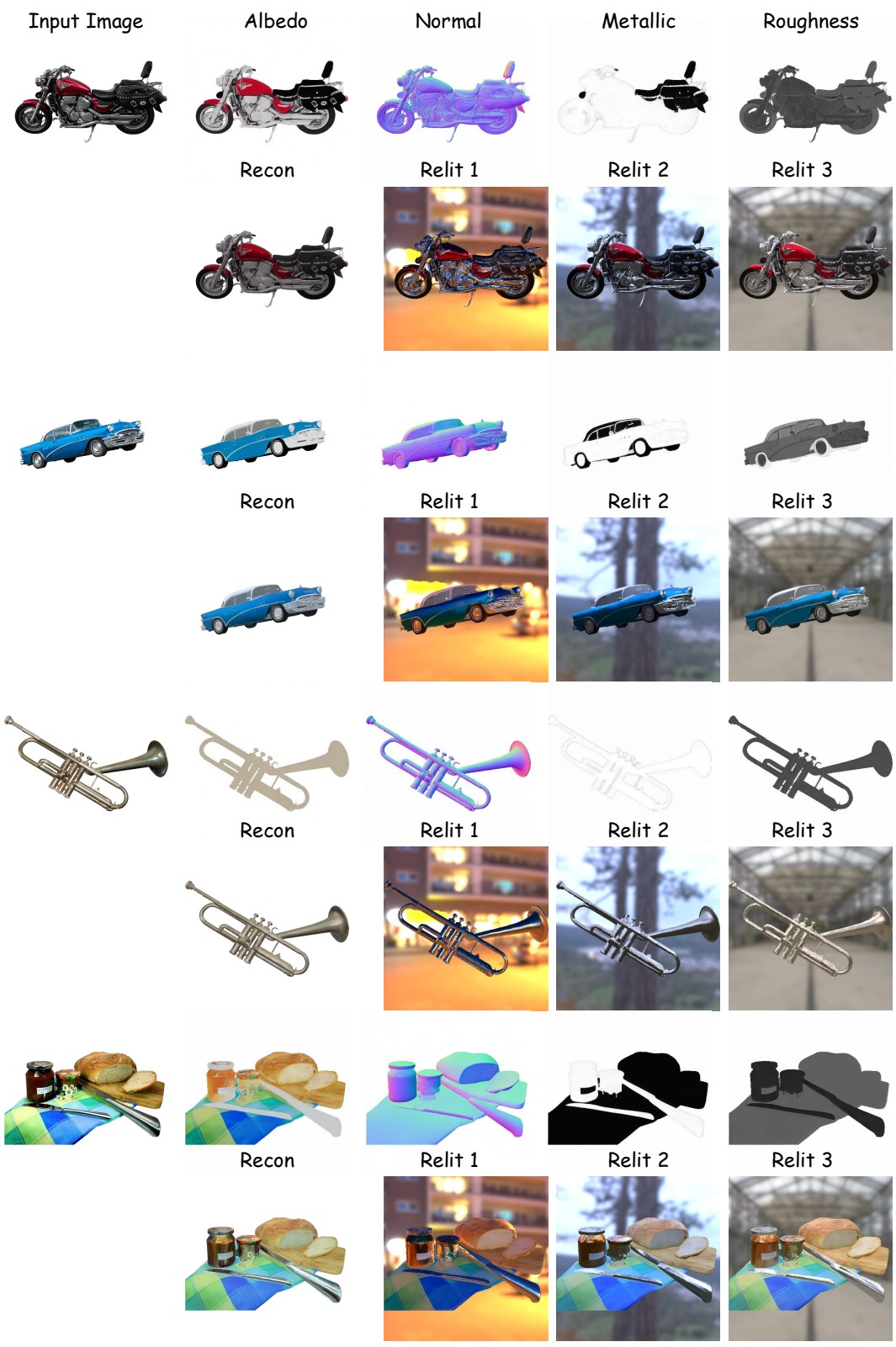

Figure 13: More results on real-world data. We also provide the reconstructed and relighting images.

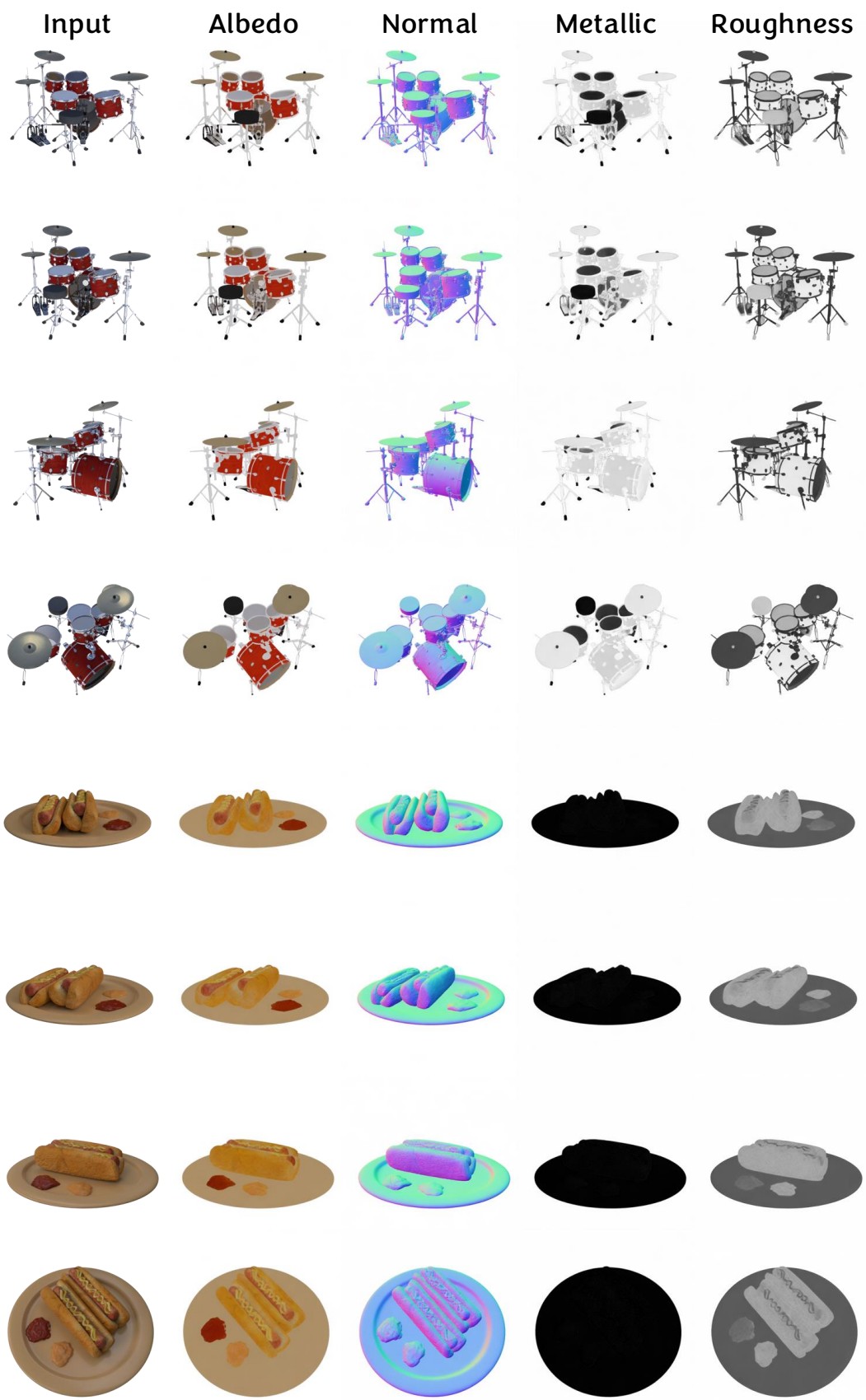

Figure 14: More results on multi-view data.

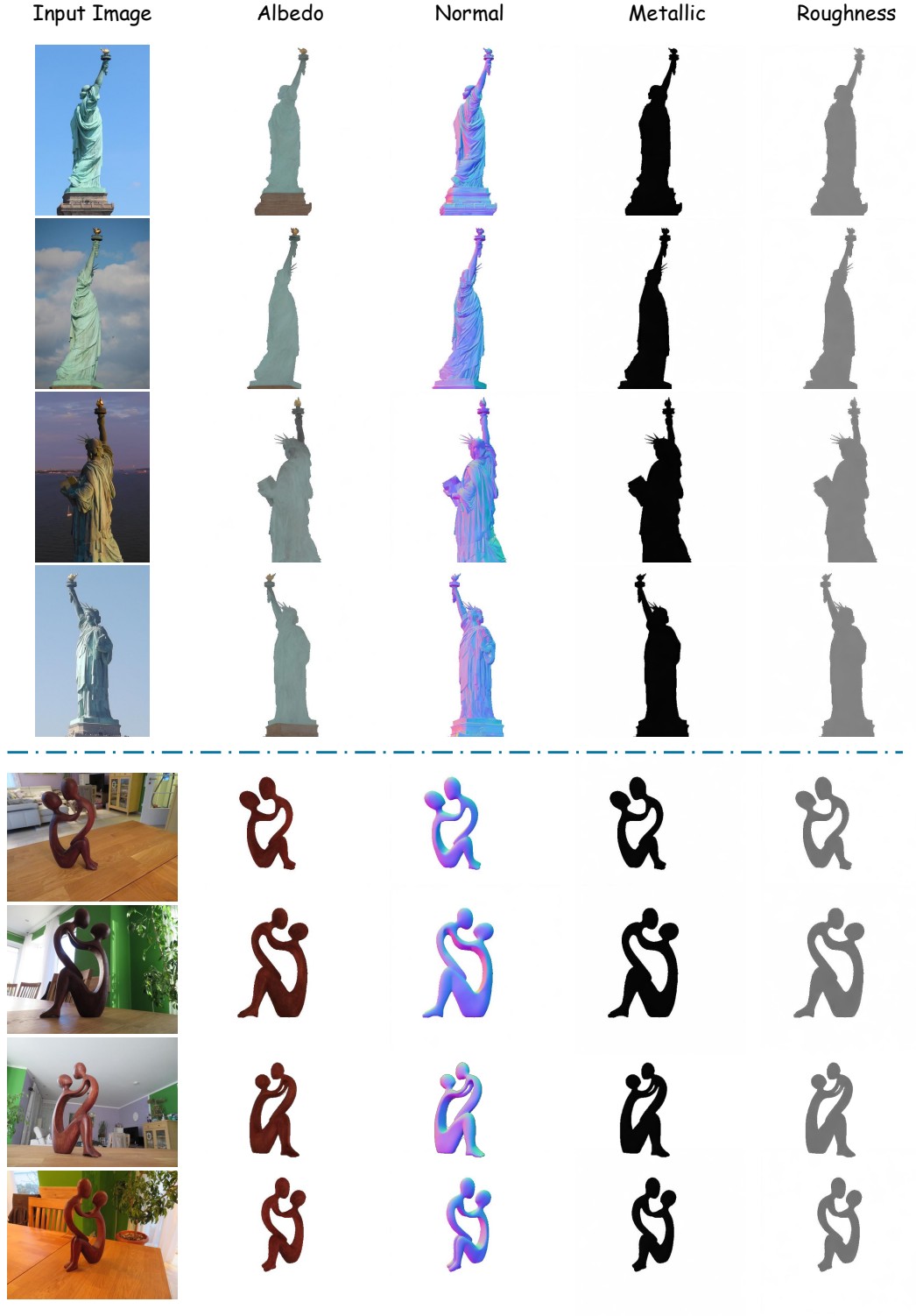

Figure 15: **Multiview images with extreme lighting variation.** For each scene in NeRD dataset (Boss et al., 2021a), we input 4 views.

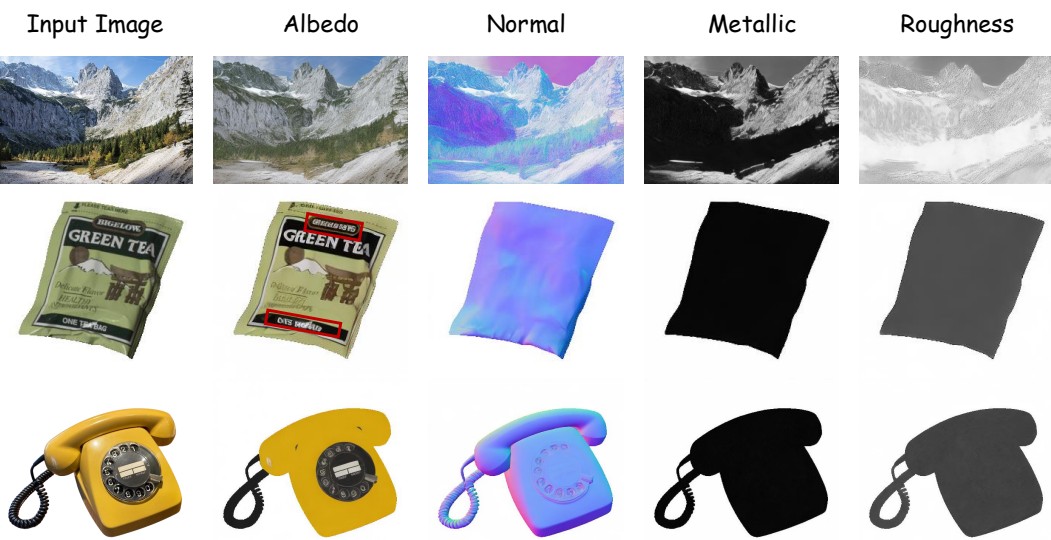

Figure 16: Failure cases.

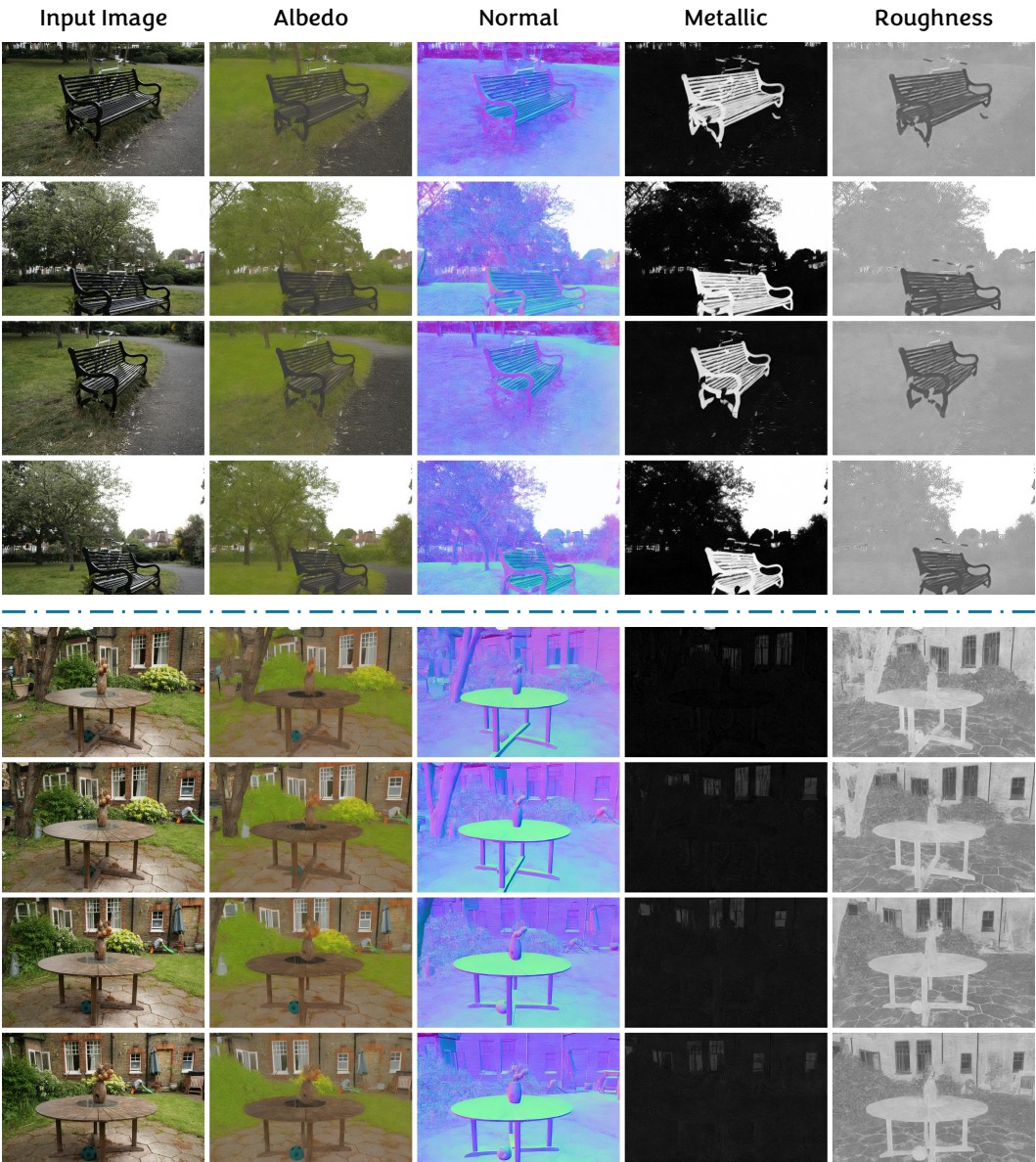

Figure 17: **Results on Mip-NeRF 360 (Barron et al., 2022) (Part 1, outdoor).** We input 4 views for each scene.

| Input Image | Albedo | Normal | Metallic | Roughness |
|---|---|---|---|---|

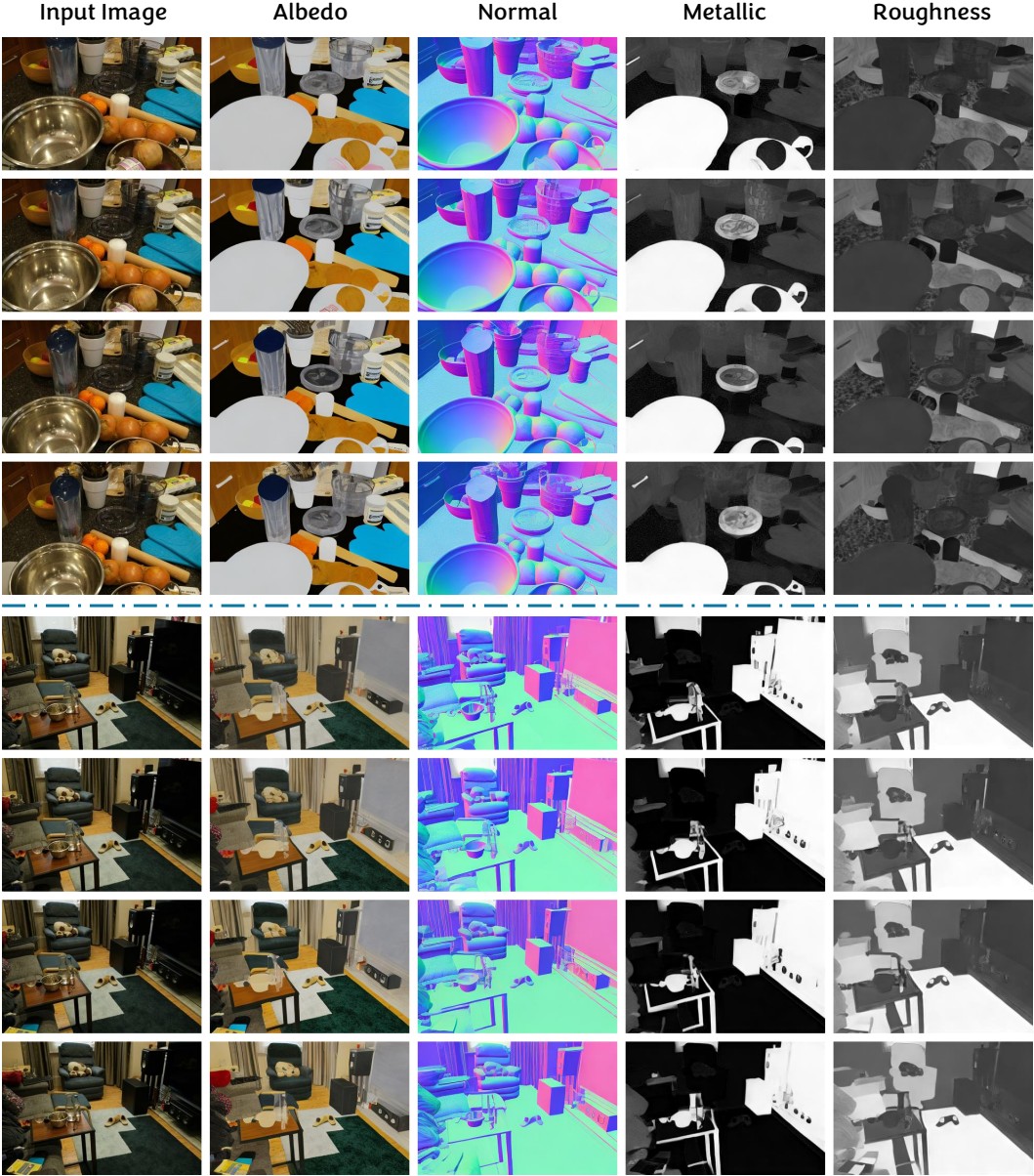

Figure 18: **Results on Mip-NeRF 360 (Barron et al., 2022) (Part 2, indoor).** We input 4 views for each scene.

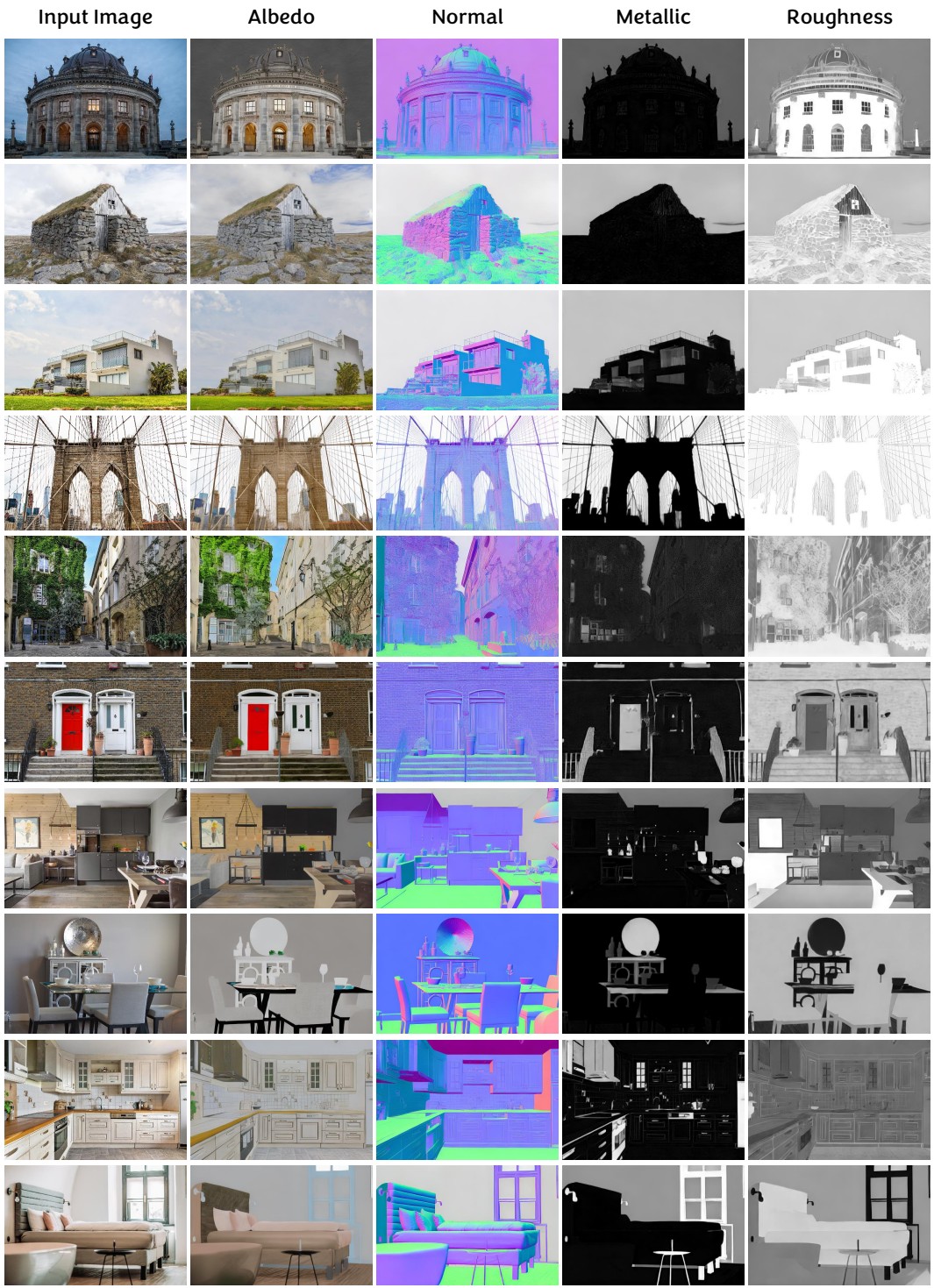

Figure 19: **Results on indoor and outdoor scenes.** Input images are collected from the Internet.

