# OpenReview forum: "IDArb: Intrinsic Decomposition for Arbitrary Number of Input Views and Illuminations"
_ICLR.cc/2025/Conference — ICLR 2025 Poster_

### Official Review · Reviewer_PUK6 · 2024-11-03

**Soundness:** 3
**Presentation:** 2
**Contribution:** 3
**Rating:** 6
**Confidence:** 4

**Summary:**

This paper proposes to utilize diffusion priors for BRDF estimation from arbitrary number of images of an object. The key technical contributions including a novel transformer that uses attention mechanism to model relations between images across different views and different intrinsic components and a new training dataset that can be used for high-quality BRDF estimation. Experiments on several popular synthetic dataset show that the proposed method achieves very impressive inverse rendering results. Moreover, those results are multi-view consistent and can be used to optimize a high-quality relightable 3D model.

**Strengths:**

1. This is a technically sound paper with clear technical contributions. The novel dataset is built in a reasonable and thoughtful way. The novel attention module can successfully capture multi-view information and multi-channel information and shown to achieve better performance on synthetic datasets.

2. From the results shown in the paper, the proposed method achieved very impressive inverse rendering results, with shadows and highlights being fully removed from the material parameters while preserving all texture details. In addition, the roughness and metallic values are piecewise smooth. It avoids the artifacts of only low roughness values near the specular highlights, which usually can observe in the optimization-based methods.

3. Given that inverse rendering is a highly ill-posed problem, it makes a lot of sense to use a generative model to sample more visually appealing and higher quality results. I believe this paper is a solid and predictable step forward compared to RGB-X paper which only considers a single image.

**Weaknesses:**

1. Experiments on real data.

My major concern of this paper is lacking real-world data experiments. I specifically recommend Stanford-ORB dataset and , where we can easily get all the comparisons with state-of-the-art dense-view reconstruction methods, including geometry and relighting accuracy. Without that experiment, it is very difficult to verify if the proposed method actually generalize well to real-world data.

2. The method section can be improved with more necessary details.

I had a little bit difficulties to under Method 3 and Figure 2. I believe I eventually understood it but felt some modifications can make it easier to read.

In Figure 2, I assume D equal to 3 and presents the number of intrinsic components. This can be explicitly explained in Figure 2. The blue, green, dark green blocks look a bit too similar and we may consider using color with stronger contrast. Also the two attention matrix has very different color and it is unclear what the colors represent. For the top half of Figure 2, it is unclear how the image latents are combined with noisy triplet latents. I assume it is a concatenation. Also for the noisy triplet latent, it may be more informative to keep some original image information so we know what it represents.

Line 246: Here, it is mentioned that each triple latent is concatenated with the Gaussian noise. Do we actually mean "added" instead of concatenated?

Illumination-augmented training: it is a bit weird to me what we sent images with different illumination to the diffusion model as during real-world data capture, we usually assume the lighting is unchanged. I wonder how this training strategy may influence the generalization on real data.

3. Quality of reference should be improved.

For the optimization-based inverse rendering, authors skipped a large number of recent works solving this problem, such as Nerfactor, NeRD, Neural-PIL, nvidffrec, nvdiffrecmc, MII, Neural-PBIR  and also many recent relightable 3D Gaussian papers. For learning-based inverse rendering, there are also many more SVBRDF estimation, lighting estimation, intrinsic decomposition papers that should be cited.

**Questions:**

Overall I like this paper and hope that it can be accepted. My major question about the paper is experiments on real data. I hope this paper can include experimental results on Stanford-ORB and Objects with lighting datasets to verify the trained model's generalization ability.

---

> ### Author Response · Authors · 2024-11-22
> **Response for reviewer PUK6**
>
> ### W-1. Experiments on Real Data
>
> Thank you for highlighting this concern. In response, we have included results on two real-world datasets: MIT-Intrinsic [1] and Stanford-ORB. Additionally, we have added StableNormal and IntrinsicNeRF for comparison, as recommended by reviewer `GJXm`.
>
> | MIT intrinsic     | SSIM $\uparrow$| PSNR $\uparrow$| LPIPS $\downarrow$|
> | ----------------- | ---------- | ---------- |---------- |
> | Ours              | 0.876 | **27.98** | **0.117** |
> | intrinsicanything | **0.896** | 25.66 | 0.150 |
>
> |Stanford-ORB    | Normal     |            | Albedo     |            |
> |-------------------|------------|------------|------------|------------|
> |                   | Cosine Distance $\downarrow$| SSIM $\uparrow$| PSNR $\uparrow$| LPIPS $\downarrow$|
> | Ours (single view input)            | $0.041$ | $0.978$ | $\underline{41.30}$ | $\underline{0.039}$ |
> | Ours (multi view input)| $\textbf{0.029}$ | $\underline{0.978}$ | $\textbf{41.46}$ | $\textbf{0.038}$ |
> | StableNormal | $\underline{0.038}$ |            |            |            |
> | IntrinsicNeRF |            | $\textbf{0.981}$ | $39.31$ | $0.048$ |
>
> We also provide qualitative results in the supplementary material: **Figure R1** of `rebuttal.pdf` (MIT-Intrinsic); `video.mp4` (Stanford-ORB).
>
> Additionally, we include results on the Internet real-world data (**Figure R6** of `rebuttal.pdf`) and the NeRD-real dataset [2] (**Figure R7** of `rebuttal.pdf` and `video.mp4`).
>
> ### W-2. Clarity and Details in Method Section
> Thanks for the constructive feedback and suggestions.
> 1. **Modifications to Figure 2:** We have update Figure 2 in response to the provided feedback:
>     1. **Clarification of D=3**: We have explicitly mentioned in the figure caption that $D=3$, which corresponds to the number of intrinsic components.
>     2. **Color Contrast**: We have adjusted the colors to provide stronger contrast, making it easier to distinguish between them.
>     3. **Attention Matrix**: We use different colors to represent the two attention matrix, as they correspond to different attention maps. We have updated the illustration of attention block to enhance understanding.
>     4. **Image Latents and Noise**: For the top half of Figure 2, we clarify that the image latents are concatenated with the noise, which is also explained in **Line 246** of the main text, where we specify that each triplet latent is **channel-concatenated** with Gaussian noise for the denoising process.
> 2.  **Illumination-augmented training:**
>     1. In real-world phenomena, lighting conditions can vary significantly due to factors like time of day, weather, or even the age of images (e.g., images of the Statue of Liberty taken over several years). This results in in-the-wild images of the same object having varying lighting conditions. Previous works [2-4] have also explored reconstruction under multiple illuminations.
>     2. By leveraging illumination-augmented training, we enhance our model's ability to handle such extreme lighting variations. This approach further allows the model to learn how to utilize photometric cues from different lighting conditions to infer intrinsic components. **Figure R7** of `rebuttal.pdf` demonstrates how our model achieves stable predictions under varying lighting conditions.
>
> ### W-3. Missing Citations
>
> Thank you for pointing this out. We have added relevant works in **Section 2.1 and 2.2** of the main paper.
>
> [1] Grosse, Roger, et al. "Ground truth dataset and baseline evaluations for intrinsic image algorithms." 2009 IEEE 12th International Conference on Computer Vision. IEEE, 2009.
>
> [2] Boss, Mark, et al. "Nerd: Neural reflectance decomposition from image collections." Proceedings of the IEEE/CVF International Conference on Computer Vision. 2021.
>
> [3] Martin-Brualla, Ricardo, et al. "Nerf in the wild: Neural radiance fields for unconstrained photo collections." Proceedings of the IEEE/CVF conference on computer vision and pattern recognition. 2021.
>
> [4] Jin, Haian, et al. "Tensoir: Tensorial inverse rendering." Proceedings of the IEEE/CVF Conference on Computer Vision and Pattern Recognition. 2023.

---

> > ### Comment · Reviewer_PUK6 · 2024-11-27
> > **Relighting results on Stanford-ORB**
> >
> > Thanks for author's response! It solves my questions about citations and minor writing issues.
> >
> > For the experiments on Stanford-ORB, one major missing experiment is the relighting experiment. Given the reconstructed BRDF map and normal map, it should not be very difficult to reconstruct the corresponding geometry mesh and BRDF texture maps and re-render the relighting results. This will allow the proposed method to be better compared to prior work. The reason albedo, roughness and metallic maps are not the best benchmark is because (1) it is hard to compared to prior works (2) the ground-truths in stanford-orb are also created through optimization so it is hard to say whether they are 100% accurate.

---

> > > ### Author Response · Authors · 2024-11-29
> > > **Re: Relighting results on Stanford-ORB**
> > >
> > > Thank you for your valuable suggestions! We fully agree that evaluating albedo using pseudo ground truth is far from ideal. we did not include an evaluation on *Novel Scene Relighting* task because Stanford-ORB uses novel lighting in **novel views** for this task. In other words, we do not have images captured from the same camera pose under different lighting conditions. Our method focuses on intrinsic component decomposition rather than novel view synthesis. This is also why Stanfor-ORB excludes learning-based methods from this benchmark, as stated: *"Since neither of [30, 2] predicts novel views or full 3D shapes, we only evaluate their depth and normal predictions and omit the other two benchmarks."*
> > >
> > > However, as suggested by reviewer `zfJQ`, re-rendering results can also serve as a metric to evaluate decomposition accuracy. Therefore, we utilized the ground truth environment maps from Stanford-ORB to re-render our decomposition results and compared them with the original images. The results are presented as follows:
> > >
> > > | | PSNR-H $\uparrow$ | PSNR-L $\uparrow$ | SSIM $\uparrow$ | LPIPS $\downarrow$ |
> > > | -------- | -------- | -------- | -------- | -------- |
> > > | Ours (single view input) | 24.11 | 31.28 | 0.969 | 0.024 |
> > > | Ours (multi view input) | 24.36 | 31.43 | 0.970 | 0.024 |

---

### Official Review · Reviewer_GJXm · 2024-11-04

**Soundness:** 3
**Presentation:** 3
**Contribution:** 3
**Rating:** 6
**Confidence:** 5

**Summary:**

This paper presents IDVI, a diffusion-based model for intrinsic decomposition that accurately estimates surface normals and material properties from multiple images under varying illuminations. It introduces a novel cross-view attention module and a new dataset, ARB-Objaverse, enabling robust training and demonstrating superior performance on various tasks, including single-image relighting and 3D reconstruction.

**Strengths:**

A new dataset is contributed.
The paper presentation is good, with figures helpful for understanding.
The results are pretty good.

**Weaknesses:**

1. The method mainly targets single objects, limiting the applications to scenes. How about performance on scenes? Could you please discuss whether the method can apply on scenes? Does it need re-training? What kind of scenes (indoor, outdoor, 360 scenes, or multi-object scenes) that it can solve, with some examples to demonstrate?
2. The method highly relies on supervised training data, while many other prior works are unsupervised/self-supervised. Could you please discuss and compare the pros/cons against self-supervised ones, and compare with them. E.g., DyNeRFactor, IntrinsicNeRF.
3. Methods to compare are too few. There are many diffusion models to estimate normal, which should also be compared. Standard benchmarks such as MIT-Intrinsic should be tested to check whether the model is overfitted on training data. E.g. StableNormal.
4. Figure 4 is compared to synthetic data, likely from the same dataset with training. That would be unfair to other methods.
I am hesitant to give a positive score mainly because with a supervised method training on a proposed synthetic dataset, it seems to have limited technical contributions. Though the results are good, experiments and evaluations could be improved.

**Questions:**

See above.

---

> ### Author Response · Authors · 2024-11-22
> **Response for reviewer GJXm**
>
> ### W-1. Performance on Scenes
>
> Thanks for the suggestion. We have evaluated our model on both indoor and outdoor scenes and present the results in **Figure R3, R4, R5** of `rebuttal.pdf` in the supplementary material. **Figure R3** and **Figure R4** show predictions on outdoor and indoor scenes from the Mip-NeRF 360 dataset [1] respectively, while **Figure R5** presents results on outdoor and indoor scenes derived from Internet images.
>
> Despite not being explicitly trained on such datasets, our model demonstrates notable generalization ability in these domains. However, the intrinsic components of objects and scenes differ significantly in their distributions. While our model exhibits generalization to scene-level data, retraining on scene-specific datasets is necessary for achieving stable and superior performance.
>
> ### W-2. Compare Pros/Cons against Unsupervised Methods
>
> Unsupervised methods, such as DyNeRFactor and IntrinsicNeRF, often require dense multi-view inputs (e.g., DyNeRFactor needs a 10-second, 30 FPS video) and can be computationally expensive (e.g., IntrinsicNeRF requires 10 hours to optimize a single scene). In contrast, supervised methods, like ours, predict intrinsic components in a feed-forward manner, which is much faster—taking only seconds or minutes.  However, supervised methods require a dataset with ground truth, which can be challenging to acquire for real-world data. When using synthetic data, these methods may face domain gaps when transferred to real-world.
>
> Supervised methods can aid unsupervised approaches by providing a reliable prior. For example, DyNeRFactor leverages a pre-trained model on the MERL dataset as a material prior to ensure that the predicted intrinsic components are physically plausible.
>
> ### W-3. Limited Comparison with Other Methods
>
> Thanks for pointing this out. We have included quantitative results on two real-world datasets:
>
> 1. MIT-Intrinsic dataset [4]: We compare our method with IntrinsicAnything on albedo prediction.
>
> | MIT intrinsic     | SSIM $\uparrow$| PSNR $\uparrow$| LPIPS $\downarrow$|
> | ----------------- | ---------- | ---------- |---------- |
> | Ours              | 0.876 | **27.98** | **0.117** |
> | intrinsicanything | **0.896** | 25.66 | 0.150 |
>
> 2. Stanford-ORB dataset [2]: We compare our method with StableNormal on normal prediction and IntrinsicNeRF on albedo prediction.
>
> |Stanford-ORB    | Normal     |            | Albedo     |            |
> |-------------------|------------|------------|------------|------------|
> |                   | Cosine Distance $\downarrow$| SSIM $\uparrow$| PSNR $\uparrow$| LPIPS $\downarrow$|
> | Ours (single view input)            | $0.041$ | $0.978$ | $\underline{41.30}$ | $\underline{0.039}$ |
> | Ours (multi view input)| $\textbf{0.029}$ | $\underline{0.978}$ | $\textbf{41.46}$ | $\textbf{0.038}$ |
> | StableNormal | $\underline{0.038}$ |            |            |            |
> | IntrinsicNeRF |            | $\textbf{0.981}$ | $39.31$ | $0.048$ |
>
> Qualitative results on MIT-Intrinsic dataset is presented in **Figure R1** of the `rebuttal.pdf` in the supplementary material. Qualitative results on Stanford-ORB dataset [2] and NeRD-real dataset [3] are in `video.mp4` in the supplimentary material.
>
> ### W-4. Additional Evaluation Results
>
> Thanks for the valuable feedback.
> 1. We understand your concern regarding the use of synthetic data for comparison. To address this, we have included additional comparisons on real-world data (as discussed in W-3). Moreover, we provide results on scene-level data (**Figure R3, R4, R5** of `rebuttal.pdf` in the supplementary material), demonstrating the strong generalization ability of our model and confirming that it is not overfitted to the training data.
> 2. Training on synthetic datasets with ground-truth supervision has been widely explored in this domain (e.g., [5-8]). Since obtaining ground-truth annotations for real-world data is challenging, some works resort to unsupervised methods. However, these approaches are often limited to specific object categories [9] or employ simplified image formation models [10] that lack physical correctness. Therefore, we render a large-scale synthetic dataset and leverage supervied training. Additionally, we argue that the domain gap between synthetic and real-world data is significantly alleviated through the use of a pre-trained diffusion model.

---

> > ### Author Response · Authors · 2024-11-22
> > **Response for reviewer GJXm (cont.)**
> >
> > [1] Barron, Jonathan T., et al. "Mip-nerf 360: Unbounded anti-aliased neural radiance fields." Proceedings of the IEEE/CVF conference on computer vision and pattern recognition. 2022.
> >
> > [2] Kuang, Zhengfei, et al. "Stanford-ORB: a real-world 3d object inverse rendering benchmark." Advances in Neural Information Processing Systems 36 (2024).
> >
> > [3] Boss, Mark, et al. "Nerd: Neural reflectance decomposition from image collections." Proceedings of the IEEE/CVF International Conference on Computer Vision. 2021.
> >
> > [4] Grosse, Roger, et al. "Ground truth dataset and baseline evaluations for intrinsic image algorithms." 2009 IEEE 12th International Conference on Computer Vision. IEEE, 2009.
> >
> > [5] Li, Zhengqin, et al. "Learning to reconstruct shape and spatially-varying reflectance from a single image." ACM Transactions on Graphics (TOG) 37.6 (2018): 1-11.
> >
> > [6] Li, Zhengqin, et al. "Inverse rendering for complex indoor scenes: Shape, spatially-varying lighting and svbrdf from a single image." Proceedings of the IEEE/CVF Conference on Computer Vision and Pattern Recognition. 2020.
> >
> > [7] Zeng, Zheng, et al. "Rgb↔ x: Image decomposition and synthesis using material-and lighting-aware diffusion models." ACM SIGGRAPH 2024 Conference Papers. 2024.
> >
> > [8] Chen, Xi, et al. "IntrinsicAnything: Learning Diffusion Priors for Inverse Rendering Under Unknown Illumination." arXiv preprint arXiv:2404.11593 (2024).
> >
> > [9] Wu, Shangzhe, et al. "De-rendering the world's revolutionary artefacts." Proceedings of the IEEE/CVF conference on computer vision and pattern recognition. 2021.
> >
> > [10] Wimbauer, Felix, Shangzhe Wu, and Christian Rupprecht. "De-rendering 3d objects in the wild." Proceedings of the IEEE/CVF Conference on Computer Vision and Pattern Recognition. 2022.

---

> > > ### Author Response · Authors · 2024-11-24
> > >
> > > We hope our responses have addressed your concerns regarding our submission. Since the discussion window closes in two days, we kindly request your feedback on our replies. If you have any additional questions or need further clarification, we would be more than happy to provide it before the window closes.

---

> > > > ### Author Response · Authors · 2024-11-26
> > > >
> > > > As the deadline for uploading the revised PDF is approaching, we kindly ask if you have any follow-up or additional questions. We would be happy to provide any additional results or clarifications to the best of our ability before the deadline. We look forward to hearing from you.

---

> > > > > ### Comment · Reviewer_GJXm · 2024-11-27
> > > > >
> > > > > Thanks for the rebuttal and further information.
> > > > >
> > > > > Although I have concerns about the method's technical contribution (supervised scheme, rendered synthetic data), I think the results and performance are good. However I think the data in Figure 5 is not ``real-world data'', maybe the authors could consider changing the descriptions.
> > > > >
> > > > > Considering the good performance, I am on the fence slightly leaning to accept. I change my score to 6.

---

> > > > > > ### Author Response · Authors · 2024-11-27
> > > > > > **Real-world data in Figure 5**
> > > > > >
> > > > > > Thank you for your valuable suggestion.
> > > > > >
> > > > > > Regarding the data in Figure 5, we used images from the Internet and made every effort to ensure they represent real-world scenarios. Below are the sources for each input image in Figure 5:
> > > > > >
> > > > > > * First row: [Gnome](https://pixabay.com/photos/troll-gnome-garden-gnome-figurine-8012851/)
> > > > > > * Second row: [Globe](https://pixers.us/wall-murals/antique-brass-armillary-sphere-on-a-wooden-stand-37203324)
> > > > > > * Third row: [Figurine](https://pixabay.com/photos/toy-figurine-japanese-anime-3731788/)
> > > > > > * Last row: [Vase](https://pixabay.com/photos/bouquet-vase-flowers-flowerpot-3175315/)
> > > > > >
> > > > > > Additionally, we have replaced the second case (globe) with [statue](https://pixabay.com/photos/statue-sculpture-bronze-angel-265421/) since the original image came from a commercial website.
> > > > > >
> > > > > > We hope this clarification helps address your concerns. Thank you again for your review!

---

### Official Review · Reviewer_cXk1 · 2024-11-04

**Soundness:** 3
**Presentation:** 2
**Contribution:** 3
**Rating:** 8
**Confidence:** 5

**Summary:**

This paper introduces IDVI, a diffusion-based multi-view model to decompose the intrinsic properties of the scanned subjective in a view-consistent manner. It leverages the commonly used Stable-Diffusion model and upgraded it with a cross-view attention and cross-domain attention to align the latents across the view/domains. The model is trained on the also proposed dataset: ARB-Objaverse, a large scale datasets rendered from Objaverse assets using multiple lighting conditions,  and achieves impressive performance in various settings.

**Strengths:**

The results from the experiments cases looks impressive and thoroughly conducted. The model can produce high fidelity normal/albedo/material decomposed that outperforms the baselines by a large margin.

The model is designed in a good shape, by leveraging the power of cross-view and cross-domain attention which are proven to be very effective in previous works.

The proposed ARB-Objaverse can largely benefit to the inverse rendering community, if it will be released to the public.

**Weaknesses:**

Issues (Ranked by priority):

The model architecture of this work is somewhat a combination of attention modules that proposed by previous works (such as GeoWizard).

Reproducibility: The model is trained on the proposed dataset ARB-Objaverse, which is rendered in a large scale. It will be hard for the community to reproduce the training if the dataset is not publicity available.

All of the experiments are conducted on object-centric cases only. While the model is trained by object data, it is also important to see how it can generalizes to other domains, for example indoor scenes (from which the RGB<->X is trained)

The Attention Block part in figure 2 is a bit confusing, a few comments that might make it more intuitive:
	(1) Add captions to explain what the images correspond to in the cross-component attention and cross-view attention (I would assume they are RGB/normal/material, and view 1/2/3 respectively)
	(2) Since the model is a combination of attention modules applied in different dimensions, the figure 2. in the paper of Group Normalization might be a good example of how to illustrate the operation in a very intuitive manner.
	(3) It is not clear how the attentions are assembled in the block. Is it follow by the order of: cross-component attention->cross-view attention->cross attention-> feed forward? If so, please clarify it in the figure.

Figure 7: Please use consistent color tones to represent good / bad performance. i.e. inverse the color map of metallic and roughness comparison.

Missing Citations:
	A couple of previous works such as NeRD, NeROIC, Neural-PIL also focuses on intrinsic decomposition on images under various illuminations; These works are also highly related to the proposed model.

**Questions:**

Overall, I think this paper is a complete work and has good performance. While it provides limited contributions on the technical part, it is a good practice of scaling up the system in the task of object intrinsic decomposition. I’m leaning to accept this paper. In addition to that, I’m happy to raise my score to a higher one if: 1) The authors will release the dataset/model and 2) The authors can show more comparisons on other domains.

---

> ### Author Response · Authors · 2024-11-22
> **Response for reviewer cXk1**
>
> ### W-1. Limited novelty
> While similar attention designs have been explored in other works such as GeoWizard, adapting such mechanisms to intrinsic decomposition is far from trivial. (1) While GeoWizard is limited to predicting single view image, our method supports arbitrary input views without requiring camera poses since our cross-view attention module effectively captures spatial relationships on its own. (2) Whereas GeoWizard employs a domain switcher to control generation across domains, we guide the process using specific text prompts via cross-attention modules. (3) Furthermore, we developed a large-scale, high-quality dataset tailored to the multi-view, varying-illumination setting. (4) Additionally, we devised a specialized training strategy to address both single-view and multi-view predictions along with a shifted noise scheduler tailored to better adapt to the material domain.
>
> ### W-2. & Q-1. Reproducibility
>
> We have made a part of the ARB-Objaverse dataset available at [this anonymous link](https://drive.google.com/file/d/1YAhYPR_i0Ij8mmFSXKP-dMh6RC2rkkfH/view?usp=sharing). We promise to release the complete dataset, along with the corresponding model and code, upon acceptance of this work.
>
> ### W-3. & Q-2. Generalization to Other Domains
>
> Thanks for pointing this out. We have evaluated our model on both indoor and outdoor scenes, with the results presented in **Figure R3, R4, R5** of `rebuttal.pdf` in the supplementary material. Despite not being explicitly trained on such datasets, our model demonstrates notable generalization ability in these domains.
>
> ### W-4. Clarifications on the Attention Block in Figure 2
>
> Thank you for your valuable feedback! We have revised **Figure 2** of the mian paper:
> 1. We added captions to explain what each image corresponds to in the cross-component attention and cross-view attention.
> 2. We refined the visualization of the attention modules to present the operations in a clearer and more intuitive manner.
> 3. We added arrows to represent the order of operations within the attention block: cross-component attention → cross-view attention → cross attention → feed-forward.
>
>
> ### W-5. Consistency in Figure 7
>
> Thank you for your suggestion. We have updated **Figure 7** of the main paper.
>
> ### W-6. Missing Citations
>
> Thank you for pointing this out. We have added relevant works such as NeRD, NeROIC, and Neural-PIL in **Section 2.1** of the main paper (Page 2, Line 105).

---

> > ### Author Response · Authors · 2024-11-24
> >
> > We hope our responses have addressed your concerns regarding our submission. Since the discussion window closes in two days, we kindly request your feedback on our replies. If you have any additional questions or need further clarification, we would be more than happy to provide it before the window closes.

---

> > > ### Author Response · Authors · 2024-11-26
> > >
> > > As the deadline for uploading the revised PDF is approaching, we kindly ask if you have any follow-up or additional questions. We would be happy to provide any additional results or clarifications to the best of our ability before the deadline. We look forward to hearing from you.

---

> ### Comment · Reviewer_cXk1 · 2024-12-01
>
> I thank the authors' effort in answering my issues and revising the paper, especially showing a demo of the datasets. I have no more questions, and am happy to raise my score. I'm looking forward to seeing the full release of the datasets in the future.

---

> > ### Author Response · Authors · 2024-12-03
> >
> > Thank you for your recognition! We are currently preparing for the release and will make our dataset and code available once the paper is accepted.

---

### Official Review · Reviewer_zfJQ · 2024-11-05

**Soundness:** 4
**Presentation:** 4
**Contribution:** 3
**Rating:** 8
**Confidence:** 4

**Summary:**

The paper proposes a method for intrinsic decomposition of object-centric images from single view or multi-view observations under varying lighting conditions, via training a diffusion model which is adapted to predicting intrinsic modalities via text prompt as switches, as well as utilizing cross attention mechanism between noised image latents of different modalities and views. The attention mechanism allows for the method to exploit dependency and consistency between intrinsic modalities and views, to achieve improved consistency and better disambiguate. The method is trained and evaluated on a new dataset rendered from Objaverse with physically-based materials and lighting, as well as real-world instances. The paper also includes ablation on model design, and applications to downstream applications.

**Strengths:**

[1] Novelty of the proposed method. The method borrows from previous methods on adapting diffusion models for intrinsic decomposition, but advances on improving the multi-view consistency by incorporating additional cross-attention mechanisms into the denosing UNet. Despite previous works which use similar attention mechanisms across multi-view inputs, the method is novel in its adaptation of the mechanism in this specific task, and is able to employ the mechanism to both cross-view and multi-task image latents.

[2] Impressive results on object centric intrinsic decomposition. The paper showcases impressive results in this task compared to some of the very recent works, demonstrating the effectiveness of the proposed attention mechanism. Moreover, special training strategy has been designed to avoid overfitting to multi-view inputs (VS single-view), and to improve quality by switching the noise scheduler.

[3] Applications to downstream applications. The paper showcases applications in downstream applications to enable relight and material editing, and application as prior for optimization-based methods.

**Weaknesses:**

[1] Limited novelty. The major novelty of the paper is in its cross-attention module, which is nodel in this formulation but not entirely new due to applications in other tasks (e.g. LRMs) by previous methods. In this case, the theoretical contribution of the attention model is limited, and the pipelines offers few other contributions besides the attention module.

[2] Limited evaluation samples. The paper offers limited account of samples in evaluation, which do not cover challenging situations including (1) scenes of high secularity (e.g. a model car); (2) multi-view images of extreme lighting variation; (3) scenes with few views. More importantly, re-rendering results are not provided as a criterion to determine the challenging modalities including roughness and metallic, which in some cases may look correct but a slight error will contribute to artifacts in re-rendered results.

[2] Evaluation on more datasets. Indoor inverse rendering has been a particular challenging domain than object-centric settings. It would be great to see evaluations on indoor setting (e.g. RGBX), if the authors are able to access sufficient indoor inverse rendering data for training.

**Questions:**

[1] Please include results of re-rendering in all of the samples using estimated intrinsic modalities, as well as more samples especially in challenging conditions.

[2] Failure cases are to be included as well.

---

> ### Author Response · Authors · 2024-11-22
> **Response for reviewer zfJQ**
>
> ### W-1. Limited novelty
> Adapting such attention mechanisms to intrinsic decomposition task is not trivial. (1) While previous works are limited to generating or predicting predefined views, our method supports arbitrary input views without requiring camera poses since our cross-view attention module effectively captures spatial relationships on its own. (2) Whereas Wonder3D and GeoWizard employ a domain switcher to control generation across domains, we guide the process using specific text prompts via cross-attention modules. (3) Furthermore, we developed a large-scale, high-quality dataset tailored to the multi-view, varying-illumination setting. (4) Additionally, we devised a specialized training strategy to address both single-view and multi-view predictions along with a shifted noise scheduler tailored to better adapt to the material domain.
>
> ### W-2. & Q-1. Limited evaluation samples and Re-rendering results
> Thank you for your valuable suggestions! We have included more challenging samples and their corresponding re-rendering results in `rebuttal.pdf` and `video.mp4` in the supplimentary material.
>
> **1. High specularity cases**
>
> We include high-specularity examples such as a motorcycle, model car, trumpet, and knife on a table in **Figure R6** of `rebuttal.pdf`. Our model effectively removes highlights and accurately identifies metallic parts, demonstrating its robustness in handling reflective surfaces.
>
> **2. Multi-view images with extreme lighting variation**
>
> We evaluated our model on three scenes from the NeRD dataset [1], captured under varying illumination at different times of day. Results for the         `StatueOfLiberty` and `MotherChild` scenes are provided in **Figure R7** of `rebuttal.pdf`, while results for the `Gnome` scene are included in `video.mp4`. Our model consistently produces plausible predictions, even under these challenging lighting conditions.
>
> **3. Scenes with few views**
>
> We tested our model with varying numbers of input views:
>
> - Single View Input: Results on real-world data are shown in **Figure R1** and **Figure R6**.
> - Four Views Input: Results are presented in **Figures R3, R4, and R7**.
> - Dense Input Views (25 views): Results are showcased in `video.mp4`.
>
> Additionally, numerical results for 1, 2, 4, and 8 input views are provided in **Figure 7** and **Tables 4–7** of the main paper.
>
> **4. Re-rendering results**
>
> We present the Re-rendering results along with relighting results of challenging cases in **Figure R6** of `rebuttal.pdf`.
>
> ### W-3. Evaluation on more datasets
>
> Thank you for your suggestion. We present qualitative results for both indoor and outdoor scenes in **Figure R3, R4, R5** of `rebuttal.pdf`. Despite not being explicitly trained on such datasets, our model demonstrates notable generalization ability in these domains.
>
> However, we argue that the intrinsic components of objects and scenes differ significantly in their distributions. While our model exhibits generalization to scene-level data, retraining on scene-specific datasets is necessary for achieving stable and superior performance. Similarly, models trained on scene datasets often experience performance degradation when applied to object-centric scenarios, as illustrated by the results of IntrinsicDiffusion in Figure 4 of the main paper.
>
> ### Q-2. Failure Cases
> Thanks for pointing this out. We provide several failure cases in **Figure R2** of `rebuttal.pdf`.
> - **Outdoor Scenes (First row).** Our model struggles with outdoor scenes, as it is primarily trained on object-centric data. While the model exhibits some generalization capability, its performance degrades in these scenarios.
> - **Textual Structures (Second Row).** When the model is faced with text, the decomposition fails to recover the correct text structures.
> - **Over-Simplified (Third Row).** In some cases, the model produces overly smooth outputs, failing to preserve subtle material details, such as the metallic features of a telephone. This issue arises from the synthetic training data, which often contains simpler material variations, leading the model to overly simplify fine-grained material properties.
>
> [1] Boss, Mark, et al. "Nerd: Neural reflectance decomposition from image collections." Proceedings of the IEEE/CVF International Conference on Computer Vision. 2021.

---

> > ### Comment · Reviewer_zfJQ · 2024-11-23
> >
> > I would like to first thank the authors for their response.
> >
> > [1] Novelty of the cross-attention modules. Paper on multi-view diffusion models have used similar modules to encourage consistency between views (e.g. MagicDrive)
> >
> > [2] Additional results. The additional results showcase consistent performance with existing results, and cover difficult or out-of-domain or failure cases with reasonable quality. I applaud the authors for taking time to work on those additional results and encourage to add those results and accompanying discussions in a later version.
> >
> > Reference:
> >
> > MagicDrice: Gao, Ruiyuan, et al. "Magicdrive: Street view generation with diverse 3d geometry control." arXiv preprint arXiv:2310.02601 (2023).

---

> > > ### Author Response · Authors · 2024-11-24
> > >
> > > Thanks for the comment. We will add those results in the final version.

---

### Official Review · Reviewer_VXK9 · 2024-11-05

**Soundness:** 2
**Presentation:** 3
**Contribution:** 2
**Rating:** 5
**Confidence:** 5

**Summary:**

This paper introduces a diffusion-based method that uses color images as input to predict material properties, including albedo, normal, metallic, and roughness. By leveraging the predicted material information and multi-view images, the method can support various applications such as relighting, material editing, photometric stereo, and inverse rendering. However, the proposed attention mechanism shows substantial similarity to prior work (Wonder3D), and the dataset lacks sufficient novelty and contribution.

**Strengths:**

1. This paper trains a diffusion model to predict material properties from color images, demonstrating strong performance compared to previous methods.

2. Extensive experiments validate the approach, showcasing its effectiveness across various applications.

3. The paper is well-organized and easy to follow.

**Weaknesses:**

1. The primary contribution of this paper is extending the original stable diffusion model from single-view, single-domain to multi-view, multi-domain processing. To accomplish this, the authors introduce a cross-view, cross-component attention mechanism. However, this attention method bears a strong resemblance to the multi-view, multi-domain attention proposed in the CVPR 2024 paper *Wonder3D*, which also extends stable diffusion to predict multiple domains. The authors have not cited or discussed *Wonder3D*, making such discussion necessary.

2. Given the similarities between the cross-view, cross-component attention in this work and *Wonder3D*, the other key contribution is a synthetic dataset designed for intrinsic image decomposition tasks. However, the dataset’s objects are sourced from Objaverse and rendered under various lighting conditions, a technique widely used in other studies. This limits the dataset’s novelty, making it challenging to consider it a significant contribution.

3. The supplementary video provides only two samples, offering limited insight. It would be highly beneficial to include additional comparisons.

**Questions:**

1. The process of 3D reconstruction from the generated multi-view images with material properties needs further elaboration. Providing more details on the reconstruction workflow, including how the material data (e.g., albedo, normal, metallic, roughness) contributes to the 3D recovery, would strengthen the clarity.

2. The functionality of the cross-view, cross-component attention mechanism also requires a more in-depth explanation. Figure 2 presents only a limited structural overview, so further details on the components and interactions within this attention scheme would greatly enhance understanding.

---

> ### Author Response · Authors · 2024-11-22
> **Response for reviewer VKX9**
>
> ### W-1. Discussion about Wonder3D
>
> Thank you for pointing this out. This is a very valuable suggestion.  Our initial inspiration came from GeoWizard, and we inadvertently overlooked that cross-domain design in GeoWizard originates from Wonder3D. We sincerely apologize for this oversight and have added the citation (**Section 2.3, Page 3, Line 144**) and the discussion (**Section 3, Page 5, Line259**) in the main paper.
>
> Although our attention block shares similarities with Wonder3D, adapting this model design to intrinsic decomposition tasks is far from trivial. (1) Unlike Wonder3D, which is limited to generating six predefined views, our method supports arbitrary input views without requiring camera poses since our cross-view attention module effectively captures spatial relationships on its own. (2) Whereas Wonder3D employs a domain switcher to control generation across domains, we guide the process using specific text prompts via cross-attention modules. (3) Furthermore, we developed a large-scale, high-quality dataset tailored to the multi-view, varying-illumination setting. (4) Additionally, we devised a specialized training strategy to address both single-view and multi-view predictions along with a shifted noise scheduler tailored to better adapt to the material domain.
>
> ### W-2. Novelty of Dataset
>
> While the technique of rendering object is indeed widely used, no existing dataset provides both diverse lighting renderings and their corresponding intrinsic components. Our dataset addresses this gap by offering 5,700,000 images of RGB rendering, albedo, normal, and material maps, generated at the cost of 1,900 GPU hours.
>
> We have uploaded part of our ARB-Objaverse dataset at [this anonymous link](https://drive.google.com/file/d/1YAhYPR_i0Ij8mmFSXKP-dMh6RC2rkkfH/view?usp=sharing) and will release the full dataset upon acceptance. As acknowledged by reviewers `cXk1`, `GJXm` and `PUK6`, we believe this contribution will be a valuable resource for the community, facilitating research in the field of inverse rendering.
>
> ### W-3. Additional Examples in the Supplementary Video
> Thank you for bringing this up. We have updated the supplementary video to include more comparisons on the Stanford-ORB[1] and NeRD-real[2] datasets. Additionally, we provide quantitative comparisons on MIT-Intrinsic[3] and Stanford-ORB[1] dataset.
>
> | MIT intrinsic     | SSIM $\uparrow$| PSNR $\uparrow$| LPIPS $\downarrow$|
> | ----------------- | ---------- | ---------- |---------- |
> | Ours              | 0.876 | **27.98** | **0.117** |
> | intrinsicanything | **0.896** | 25.66 | 0.150 |
>
> |Stanford-ORB    | Normal     |            | Albedo     |            |
> |-------------------|------------|------------|------------|------------|
> |                   | Cosine Distance $\downarrow$| SSIM $\uparrow$| PSNR $\uparrow$| LPIPS $\downarrow$|
> | Ours (single view input)            | $0.041$ | $0.978$ | $\underline{41.30}$ | $\underline{0.039}$ |
> | Ours (multi view input)| $\textbf{0.029}$ | $\underline{0.978}$ | $\textbf{41.46}$ | $\textbf{0.038}$ |
> | StableNormal [4] | $\underline{0.038}$ |            |            |            |
> | IntrinsicNeRF [5]|            | $\textbf{0.981}$ | $39.31$ | $0.048$ |
>
> ### Q-1. Elaboration on 3D Reconstruction
> Thank you for your suggestion. We have revised this section in the main paper (**Page 10, Line 514**) to provide additional details. Rather than directly reconstructing the 3D shape from the predicted intrinsic maps, we use the predicted intrinsic components as material priors to enhance existing 3D reconstruction methods. Specifically, we select NVDiffRecMC [6] to validate the effectiveness of our priors.
> During each iteration, we introduce an additional L2 loss between the intrinsic components reconstructed by NVDiffRecMC and those predicted by our method. This significantly alleviates color-shifting issues in albedo reconstruction and ensures the physical plausibility of the reconstructed materials.
>
> ### Q-2. Explanation of the Cross-View, Cross-Component Attention Mechanism
> Thanks for the suggestion. We have updated Figure 2: Overview of IDVI and its caption (**Page 4, Line 182)** to improve clarity.
>
> For cross-component attention, we extend the original self-attention module to operate across different components. Specifically, the input tensor is reshaped from $[(B,V,D),H,W,C]$ to $[(B,V),(D,HW),C]$ before attention calculation, where $B$, $V$ and $D$ represent batch size, number of views and number of intrinsic components.
>
> For cross-view attention, the input is reshaped from $[(B,V,D),H,W,C]$ to $[(B,D),(V,HW),C]$ before applying self-attention.

---

> > ### Author Response · Authors · 2024-11-22
> > **Response for reviewer VKX9 (cont.)**
> >
> > [1] Kuang, Zhengfei, et al. "Stanford-ORB: a real-world 3d object inverse rendering benchmark." Advances in Neural Information Processing Systems 36 (2024).
> >
> > [2] Boss, Mark, et al. "Nerd: Neural reflectance decomposition from image collections." Proceedings of the IEEE/CVF International Conference on Computer Vision. 2021.
> >
> > [3] Grosse, Roger, et al. "Ground truth dataset and baseline evaluations for intrinsic image algorithms." 2009 IEEE 12th International Conference on Computer Vision. IEEE, 2009.
> >
> > [4] Ye, Chongjie, et al. "Stablenormal: Reducing diffusion variance for stable and sharp normal." arXiv preprint arXiv:2406.16864 (2024).
> >
> > [5] Ye, Weicai, et al. "Intrinsicnerf: Learning intrinsic neural radiance fields for editable novel view synthesis." Proceedings of the IEEE/CVF International Conference on Computer Vision. 2023.
> >
> > [6] Hasselgren, Jon, Nikolai Hofmann, and Jacob Munkberg. "Shape, light, and material decomposition from images using monte carlo rendering and denoising." Advances in Neural Information Processing Systems 35 (2022): 22856-22869.

---

> > ### Comment · Reviewer_VXK9 · 2024-11-22
> >
> > Thanks for your response. I have still some questions.
> >
> > **W-1. Discussion about Wonder3D**
> >
> > q1: The authors explains that "Unlike Wonder3D, which is limited to generating six predefined views, our method supports arbitrary input views..". Does this method could produce arbitrary views or predefined views? If this method still produces predefined views, the only difference with wonder3D is the input, but the cross-domain cross-view attention is highly similar with Wonder3D.
> >
> > q2. The author claims that “Whereas Wonder3D employs a domain switcher to control generation across domains, we guide the process using specific text prompts via cross-attention modules.” However, Geowizard also uses specific text prompts via cross-attention, such operation has already been used in geowizard.
> >
> > **Q-2. Explanation of the Cross-View, Cross-Component Attention Mechanism**
> >
> > Based on the explanation, the idea of such attention mechanism is nearly the same with cross-domain multi-view attention proposed in Wonder3D. I acknowledge the good results and contributions in terms of intrinsic decomposition, but I believe such attention operation is over-claimed. If the authors could reconsider the claimed contributions, I am willing to raise my score.

---

> > > ### Author Response · Authors · 2024-11-22
> > >
> > > Thank you for your valuable feedback.
> > >
> > > **W-1-Q-1**
> > >
> > > To clarify, our method is not a generation approach but designed for prediction tasks in a generative way. Specifically, our method focuses on intrinsic decomposition rather than novel RGB view or intrinsic component synthesis. Therefore, the produced views correspond directly to the input views but are transformed into the intrinsic domain.
> > >
> > > By "arbitrary input views," we mean that our model can process input views of varying number and viewpoints. While trained on single and three-view inputs, the model generalizes well to any arbitrary input settings. We apologize for the misleading phrasing in our earlier response.
> > >
> > >
> > > **W-1-Q-2**
> > >
> > > Upon reviewing the paper and code for GeoWizard, we found that the paper describes the geometry switcher and scene decoupler as one-dimensional vectors. However, it released a v2 model in GitHub, which uses text embeddings to indicate different scenes.
> > >
> > > In light of this, we have removed the statement *"Unlike Wonder3D and GeoWizard, which employ a domain switcher to control generation across domains, we guide the process using specific text prompts via cross-attention modules."* from main paper and will no longer claim this as a difference.
> > >
> > > **Q-2**
> > >
> > > Thank you for your suggestion. We have revised our paper and re-claimed our contribution as ***adopt the cross-view, cross-component attention module from Wonder3D*** (Page 2, Line 082).
> > >
> > > We sincerely apologize again for this oversight and appreciate your constructive feedback. Please let us know if there are further points you'd like us to consider.

---

> > > > ### Author Response · Authors · 2024-11-24
> > > >
> > > > We hope our responses have addressed your concerns regarding our submission. Since the discussion window closes in two days, we kindly request your feedback on our replies. If you have any additional questions or need further clarification, we would be more than happy to provide it before the window closes.

---

> > > > > ### Author Response · Authors · 2024-11-26
> > > > >
> > > > > As the deadline for uploading the revised PDF is approaching, we kindly ask if you have any follow-up or additional questions. We would be happy to provide any additional results or clarifications to the best of our ability before the deadline. We look forward to hearing from you.

---

> > > > > > ### Comment · Reviewer_VXK9 · 2024-11-29
> > > > > >
> > > > > > Thanks for the updating. Considering the authors revised the claims, I raised the score.

---

> > > > > > > ### Author Response · Authors · 2024-11-29
> > > > > > >
> > > > > > > Thank you for your comment. We're pleased to hear that the revisions have addressed your concerns. If you have any further questions, we would be more than happy to assist.

---

### Author Response · Authors · 2024-11-21
**General response for all reviewers**

We sincerely thank the reviewers for their constructive feedback and valuable suggestions. We are glad that the reviewers recognized the strong performance of our method (`VXK9`, `zfJQ`, `cXk1`, `PUK6`) and acknowledged the contribution of our constructed dataset for the community (`cXk1`, `GJXm`, `PUK6`). In response to the feedback, we have made several improvements to the main paper (marked in blue):
- We added more relevant works on optimization-based and learning-based methods in **Section 2.1 and 2.2**.
- We included a discussion on Wonder3D and GeoWizard in **Section 3, Page 5, Line259**
- We updated **Figure 2** and its caption to enhance clarity. We also explicitly mentioned that triplet latent is channel-concatenated with Gaussian noise for clarity (**Page 5, Line 246**).
- We revised **Figure 7** to use consistent color tones.
- We elaborated on how our method serves as a prior to enhance optimization-based inverse rendering methods (**Page10, Line 513**).

We also expanded the supplementary material to include more results:
* `video.mp4`: Results on NeRF, Stanford-ORB and NeRD-real datasets.
* `rebuttal.pdf`:
  * **Figure R1**: Qualitative comparisons on MIT-Intrinsic dataset.
  * **Figure R2**: Failure cases.
  * **Figure R3**: Results on outdoor scenes from Mip-NeRF 360 dataset.
  * **Figure R4**: Results on indoor scenes from Mip-NeRF 360 dataset.
  * **Figure R5**: Results on outdoor and indoor scenes from Internet images.
  * **Figure R6**: Results on high-specularity cases, including corresponding re-rendering and relighting results.
  * **Figure R7**: Results on multiview images under varying illuminations from NeRD-real dataset

We provide quantitative comparisons on the MIT-Intrinsic and Stanford-ORB datasets as follows:
| MIT intrinsic     | SSIM $\uparrow$| PSNR $\uparrow$| LPIPS $\downarrow$|
| ----------------- | ---------- | ---------- |---------- |
| Ours              | 0.876 | **27.98** | **0.117** |
| intrinsicanything | **0.896** | 25.66 | 0.150 |

|Stanford-ORB    | Normal     |            | Albedo     |            |
|-------------------|------------|------------|------------|------------|
|                   | Cosine Distance $\downarrow$| SSIM $\uparrow$| PSNR $\uparrow$| LPIPS $\downarrow$|
| Ours (single view input)            | $0.041$ | $0.978$ | $\underline{41.30}$ | $\underline{0.039}$ |
| Ours (multi view input)| $\textbf{0.029}$ | $\underline{0.978}$ | $\textbf{41.46}$ | $\textbf{0.038}$ |
| StableNormal | $\underline{0.038}$ |            |            |            |
| IntrinsicNeRF |            | $\textbf{0.981}$ | $39.31$ | $0.048$ |

---

### Meta-Review · Area_Chair_Cfey · 2024-12-22

**Metareview:**

The paper introduces IDVI, a diffusion-based model for intrinsic decomposition that uses cross-view attention for consistent estimation of surface normals and material properties under varying illuminations. Trained on the new ARB-Objaverse dataset, it achieves state-of-the-art performance and supports tasks like relighting and 3D reconstruction.

Strengths:
Novel Method: Introduces cross-view and cross-domain attention for improved multi-view consistency in intrinsic decomposition.
Impressive Results: Achieves high-quality decompositions, outperforming baselines while preserving texture and removing artifacts.
Broad Applications: Supports tasks like relighting, material editing, and inverse rendering.
Dataset Contribution: Provides ARB-Objaverse, a valuable resource for the inverse rendering community.

Weaknesses:
Limited Novelty: The cross-view and cross-domain attention mechanism shows similarities to prior works (e.g., Wonder3D, GeoWizard), with insufficient citations and discussion on its contributions compared to existing approaches.
Generalization Concerns: Experiments are limited to object-centric synthetic data, with no evaluation on real-world datasets (e.g., Stanford-ORB) or challenging scenarios like indoor scenes or scenes with extreme lighting variation.
Dataset Dependency: Heavy reliance on the synthetic ARB-Objaverse dataset raises concerns about overfitting, with no experiments on standard benchmarks (e.g., MIT-Intrinsic) or unsupervised/self-supervised comparisons to assess broader applicability.
Insufficient Evaluation: Missing key comparisons with related diffusion and inverse rendering methods, as well as a lack of challenging or diverse visual examples, limits the robustness and fairness of the evaluations.

The paper receives mixed positive and negative reviews.
We are (weakly) accepting the paper and want the authors to improve their camera-ready paper further based on the reviews, especially the negative ones.

**Additional Comments On Reviewer Discussion:**

The authors made significant revisions, addressing key concerns:
Expanded Related Works: Added discussions on Wonder3D, GeoWizard, and other relevant methods.
Improved Clarity: Enhanced explanations in Figure 2 and standardized Figure 7’s color tones.
Real-World Validation: Provided quantitative results on MIT-Intrinsic and Stanford-ORB datasets, improving generalization evidence.
Supplementary Updates: Added extensive new results, covering diverse datasets and scenarios.

Reviewer VXK9 raised their score slightly but remained negative, citing concerns about technical contributions and reliance on synthetic data. Reviewer GJXm, while noting similar concerns, acknowledged the strong results and raised their score to a positive (6), leaning toward acceptance. Reviewer cXk1 expressed satisfaction with the revisions and dataset demo, raising their score and looking forward to the dataset release.

The paper's novelty got challenged, but got positive scores from the majority. We hence weakly accept the paper, and would like the authors to improve their camera-ready paper based on all reviews.

---

### Decision · Program_Chairs · 2025-01-22

Accept (Poster)